# DON'T USE LARGE MINI-BATCHES, USE LOCAL SGD

**Tao Lin**
EPFL, Switzerland
tao.lin@epfl.ch

**Sebastian U. Stich**
EPFL, Switzerland
sebastian.stich@epfl.ch

**Kumar Kshitij Patel**
IIT Kanpur, India
kumarkshitijpatel@gmail.com

**Martin Jaggi**
EPFL, Switzerland
martin.jaggi@epfl.ch

## ABSTRACT

Mini-batch stochastic gradient methods (SGD) are state of the art for distributed training of deep neural networks. Drastic increases in the mini-batch sizes have lead to key efficiency and scalability gains in recent years. However, progress faces a major roadblock, as models trained with large batches often do not generalize well, i.e. they do not show good accuracy on new data.

As a remedy, we propose a *post-local* SGD and show that it significantly improves the generalization performance compared to large-batch training on standard benchmarks while enjoying the same efficiency (time-to-accuracy) and scalability. We further provide an extensive study of the communication efficiency vs. performance trade-offs associated with a host of *local SGD* variants.

## 1 INTRODUCTION

Fast and efficient training of large scale deep-learning models relies on distributed hardware and on distributed optimization algorithms. For efficient use of system resources, these algorithms crucially must (i) enable parallelization while being communication efficient, and (ii) exhibit good generalization behaviour, i.e. good performance on unseen data (test-set). Most machine learning applications currently depend on stochastic gradient descent (SGD) (Robbins & Monro, 1985) and in particular its mini-batch variant (Bottou, 2010; Dekel et al., 2012). However, this algorithm faces generalization difficulties in the regime of very large batch sizes as we will review now.

**Mini-batch SGD.** For a sum-structured optimization problem of the form $\min_{\boldsymbol{w}\in\mathbb{R}^d} \frac{1}{N}\sum_{i=1}^N f_i(\boldsymbol{w})$ where $\boldsymbol{w} \in \mathbb{R}^d$ denotes the parameters of the model and $f_i\colon \mathbb{R}^d \to \mathbb{R}$ the loss function of the $i$-th training example, the mini-batch SGD update for $K \geqslant 1$ workers is given as

$$\boldsymbol{w}_{(t+1)} := \boldsymbol{w}_{(t)} - \gamma_{(t)}\left[\tfrac{1}{K}\sum_{k=1}^K \; \tfrac{1}{B}\sum_{i\in\mathcal{I}_{(t)}^k}\nabla f_i\big(\boldsymbol{w}_{(t)}\big)\right], \tag{1}$$

where $\gamma_{(t)} > 0$ denotes the learning rate and $\mathcal{I}_{(t)}^k \subseteq [N]$ the subset (mini-batch) of training datapoints selected by worker $k$ (typically selected uniformly at random from the locally available datapoints on worker $k$). For convenience, we will assume the same batch size $B$ per worker.

**Local SGD.** Motivated to better balance the available system resources (computation vs. communication), local SGD (a.k.a. local-update SGD, parallel SGD, or federated averaging) has recently attracted increased research interest (Mcdonald et al., 2009; Zinkevich et al., 2010; McDonald et al., 2010; Zhang et al., 2014; 2016; McMahan et al., 2017). In local SGD, each worker $k \in [K]$ evolves a local model by performing $H$ sequential SGD updates with mini-batch size $B_{\text{loc}}$, before communication (synchronization by averaging) among the workers. Formally,

$$\boldsymbol{w}_{(t)+h+1}^k := \boldsymbol{w}_{(t)+h}^k - \gamma_{(t)}\left[\tfrac{1}{B_{\text{loc}}}\sum_{i\in\mathcal{I}_{(t)+h}^k}\nabla f_i\big(\boldsymbol{w}_{(t)+h}^k\big)\right], \quad \boldsymbol{w}_{(t+1)}^k := \tfrac{1}{K}\sum_{k=1}^K \boldsymbol{w}_{(t)+H}^k, \tag{2}$$

where $\boldsymbol{w}_{(t)+h}^k$ denotes the local model on machine $k$ after $t$ global synchronization rounds and subsequent $h \in [H]$ local steps ($\mathcal{I}_{(t)+h}^k$ is defined analogously). Mini-batch SGD is a special case

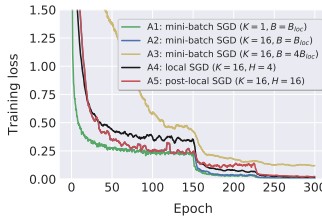
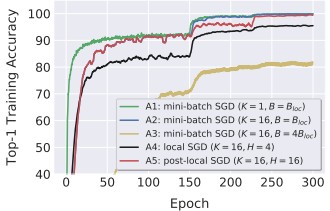
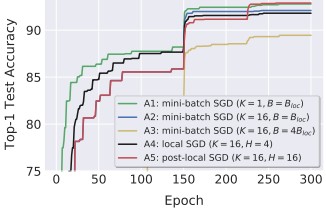

| (a) Training loss. | (b) Training top-1 accuracy. | (c) Test top-1 accuracy. |

|  | accuracy on training set | | accuracy on test set | | system performance | | |
|---|---|---|---|---|---|---|---|
| **Algorithm** | loss value | top-1 acc. | | top-1 acc. | | parallelism | communication | |
| A1: small mini-batch SGD ($K=1, B=B_{loc}$) | 0.01 | 100% | *excellent* | 93% | *excellent* | ×1 | - | *poor* |
| A2: large mini-batch SGD ($K=16, B=B_{loc}$) | 0.01 | 100% | *excellent* | 92% | *good* | ×16 | ÷1 | *ok* |
| A3: huge mini-batch SGD ($K=16, B=4B_{loc}$) | 0.10 | 81% | *poor* | 89% | *poor* | ×16 | ÷4 | *good* |
| A4: local SGD ($K=16, H=4$) | 0.01 | 95% | *ok* | 92% | *good* | ×16 | ÷4 | *good* |
| A5: post-local SGD ($K=16, H=16$) | 0.01 | 99% | *excellent* | 93% | *excellent* | ×16 | ÷1 (phase 1), ÷16 (phase 2) | *good* |

Figure 1: Illustration of the **generalization gap**. Large-batch SGD (*A2, blue*) matches the training curves of small-batch SGD (*A1, green*), i.e. has no optimization difficulty (left & middle). However, it does not reach the same test accuracy (right) while the proposed post-local SGD (*A5, red*) does. Post-local SGD (*A5*) is defined by starting local SGD from the model obtained by large-batch SGD (*A2*) at epoch 150. Mini-batch SGD with larger mini-batch size (*A3, yellow*) even suffers from optimization issues. Experiments are for ResNet-20 on CIFAR-10 ($B_{loc} = 128$), with fined-tuned learning rate for mini-batch SGD with the warmup scheme in Goyal et al. (2017). The inline table highlights the comparison of system/generalization performance for different algorithms.

if $H = 1$ and $B_{loc} = B$. Furthermore, the communication patterns of both algorithms are identical if $B = HB_{loc}$. However, as the updates (eq. (2)) are different from the mini-batch updates (eq. (1)) for any $H > 1$, the generalization behavior (test error) of both algorithms is expected to be different.

**Large batch SGD.** Recent schemes for scaling training to a large number of workers rely on standard mini-batch SGD (1) with very large overall batch sizes (Shallue et al., 2018; You et al., 2018; Goyal et al., 2017), i.e. increasing the global batch size linearly with the number of workers $K$. However, the batch size interacts differently with the overall system efficiency on one hand, and with generalization performance on the other hand. While a larger batch size in general increases throughput, it may negatively affect the final accuracy on both the train- and test-set. Two main scenarios of particular interest can be decoupled as follows:

**Scenario 1. The communication restricted setting**, where the synchronization time is much higher than the gradient computation time. In this case the batch size of best efficiency is typically large, and is achieved by using partial computation (gradient accumulation) while waiting for communication. We in particular study the interesting case when the mini-batch sizes of both algorithms satisfy the relation $B = HB_{loc}$, as in this case both algorithms (local SGD and mini-batch SGD) evaluate the same number of stochastic gradients between synchronization steps.

**Scenario 2. The regime of poor generalization of large-batch SGD**, that is the use of very large overall batches (often a significant fraction of the training set size), which is known to cause drastically decreased generalization performance (Chen & Huo, 2016; Keskar et al., 2017; Hoffer et al., 2017; Shallue et al., 2018; Golmant et al., 2018). If sticking to standard mini-batch SGD and maintaining the level of parallelisation, the batch size $B$ would have to be reduced below the device (locally optimal) capacity in order to alleviate this generalization issue, which however impacts training time[1].

**Main Results.** Key aspects of the empirical performance of local SGD compared to mini-batch baselines are illustrated in Figure 1. In scenario 1), comparing local SGD with $H = 4$ (A4) with mini-batch SGD of same effective batch size $B = 4B_{loc}$ (A3) reveals a stark difference, both in terms of train and test error (local SGD achieves lower training loss and higher test accuracy). This motivates the use of *local SGD as an alternative to large-batch training*—a hypothesis that we confirm in our experiments. Further, in scenario 2), mini-batch SGD with smaller batch size $B = B_{loc}$ (A2) is observed to suffer from poor generalization, although the training curve matches the single-

---

[1]Note that in terms of efficiency on current GPUs, the computation time on device for small batch sizes is not constant but scales non-linearly with $B$, as shown in Table 7 in Appendix A.

machine baseline (A1). *The generalization gap can thus not be explained as an optimization issue alone*. Our proposed *post-local SGD* (A5) (defined by starting local SGD from the model obtained by large-batch SGD (*A2*) at epoch $150$) closes this generalization gap with the single-machine baseline (A1) and is also more communication efficient than the mini-batch competitors. In direct comparison, post-local SGD is more communication-efficient than mini-batch SGD (while less than local SGD). It achieves better generalization performance than both these algorithms.

**Contributions.** Our main contributions can thus be summarized as follows:

- **Trade-offs in Local SGD:** We provide the first comprehensive empirically study of the trade-offs in local SGD for deep learning—when varying the number of workers $K$, number of local steps $H$ and mini-batch sizes—for both scenarios 1) on communication efficiency and 2) on generalization.
- **Post-local SGD:** We propose *post-local SGD*, a simple but very efficient training scheme to address the current generalization issue of large-batch training. It allows us to scale the training to much higher number of parallel devices. Large batches trained by post-local SGD enjoy improved communication efficiency, while at the same time strongly outperforming most competing small and large batch baselines in terms of accuracy. Our empirical experiments on standard benchmarks show that post-local SGD can reach flatter minima than large-batch SGD on those problems.

## 2 RELATED WORK

**The generalization gap in large-batch training.** State-of-the-art distributed deep learning frameworks (Abadi et al., 2016; Paszke et al., 2017; Seide & Agarwal, 2016) resort to synchronized large-batch SGD training, allowing scaling by adding more computational units and performing data-parallel synchronous SGD with mini-batches divided between devices. Training with large batch size (e.g. batch size $> 10^3$ on ImageNet) typically degrades the performance both in terms of training and test error (often denoted as *generalization gap*) (Chen & Huo, 2016; Li, 2017; Li et al., 2014; Keskar et al., 2017; Shallue et al., 2018; McCandlish et al., 2018; Golmant et al., 2018; Masters & Luschi, 2018). Goyal et al. (2017) argue that the test error degrades because of optimization issues and propose to use a "learning rate warm-up" phase with linear scaling of the step-size. You et al. (2017a) propose Layer-wise Adaptive Rate Scaling (LARS) to scale to larger mini-batch size, but the generalization gap does not vanish. Hoffer et al. (2017) argue that the generalization gap can be closed when increasing the number of iterations along with the batch size. However, this diminishes the efficiency gains of parallel training.

Keskar et al. (2017) empirically show that larger batch sizes correlate with sharper minima (Hochreiter & Schmidhuber, 1997) found by SGD and that flat minima are preferred for better generalization. This interpretation—despite being debated in Dinh et al. (2017)—was further developed in Hoffer et al. (2017); Yao et al. (2018); Izmailov et al. (2018). Neelakantan et al. (2015) propose to add isotropic white noise to the gradients to avoid over-fitting and better optimization. Zhu et al. (2019); Xing et al. (2018) further analyze "structured" anisotropic noise and highlight the importance of anisotropic noise (over isotropic noise) for improving generalization. The importance of the scale of the noise for non-convex optimization has also been studied in Smith & Le (2018); Chaudhari & Soatto (2018). Wen et al. (2019) propose to inject noise (sampled from the expensive empirical Fisher matrix) to large-batch SGD. However, to our best knowledge, none of the prior work (except our post-local SGD) can provide a computation efficient way to inject noise to achieve as good generalization performance as small-batch SGD, for both of CIFAR and ImageNet experiments.

**Local SGD and convergence theory.** While mini-batch SGD is very well studied (Zinkevich et al., 2010; Dekel et al., 2012; Takáč et al., 2013), the theoretical foundations of local SGD variants are still developing. Jain et al. (2018) study one-shot averaging on quadratic functions and Bijral et al. (2016) study local SGD in the setting of a general graph of workers. A main research question is whether local SGD provides a linear speedup with respect to the number of workers $K$, similar to mini-batch SGD. Recent work partially confirms this, under the assumption that $H$ is not too large compared to the total iterations $T$. Stich (2019) and Patel & Dieuleveut (2019) show convergence at rate $\mathcal{O}\big((KTHB_{\text{loc}})^{-1}\big)$ on strongly convex and smooth objective functions when $H = \mathcal{O}(T^{1/2})$. For smooth non-convex objective functions, Zhou & Cong (2018) show a rate of $\mathcal{O}\big((KTB_{\text{loc}})^{-1/2}\big)$ (for the decrement of the stochastic gradient), Yu et al. (2019) give an improved

result $\mathcal{O}\big((HKTB_{\mathrm{loc}})^{-1/2}\big)$ when $H = \mathcal{O}(T^{1/4})$. For a discussion of more recent theoretical results we refer to (Kairouz et al., 2019). Alistarh et al. (2018) study convergence under adversarial delays. Zhang et al. (2016) empirically study the effect of the averaging frequency on the quality of the solution for some problem cases and observe that more frequent averaging at the beginning of the optimization can help. Similarly, Bijral et al. (2016) argue to average more frequently at the beginning.

## 3 POST-LOCAL SGD AND HIERARCHICAL LOCAL SGD

In this section we present two novel variants of local SGD. First, we propose post-local SGD to reach high generalization accuracy (cf. Section 4.2 below), and second, hierarchical SGD designed from a systems perspective aiming at optimal resource adaptivity (computation vs. communication trade-off).

**Post-local SGD: Large-batch Training Alternative for Better Generalization.** We propose post-local SGD, a variant where local SGD is only started in the second phase of training, after $t'$ initial steps[2] with standard mini-batch SGD. Formally, the update in (2) is performed with a iteration dependent $H_{(t)}$ given as

$$H_{(t)} = \begin{cases} 1, & \text{if } t \leqslant t', \quad \textit{(mini-batch SGD)} \\ H, & \text{if } t > t'. \quad \textit{(local SGD)} \end{cases} \qquad \text{(post-local SGD)}$$

As the proposed scheme is identical to mini-batch SGD in the first phase (with local batch size $B = B_{\mathrm{loc}}$), we can leverage previously tuned learning rate warm-up strategies and schedules for large-batch training (Goyal et al., 2017) without additional tuning. Note that we only use 'small' local mini-batches of size $B = B_{\mathrm{loc}}$ in the warm-up phase, and switch to the communication efficient larger effective batches $HB_{\mathrm{loc}}$ in the second phase, while also achieving better generalization in our experiments (cf. also the discussion in Section 5). We would like to point out that it is crucial to use local SGD in the second phase, as e.g. just resorting to large batch training does achieve worse performance (see e.g. 3rd row in Table 2 below).

**Hierarchical Local SGD: Optimal Use of Systems Resources in Heterogeneous Systems.** Real world systems come with different communication bandwidths on several levels, e.g. with GPUs or other accelerators grouped hierarchically within a chip, machine, rack or even at the level of entire data-centers. In this scenario, we propose to employ local SGD as an inner loop on each level of the hierarchy, adapted to the corresponding computation vs communication trade-off of that particular level. The resulting scheme, *hierarchical local SGD*, can offer significant benefits in terms of system adaptivity and performance, as we show with experiments and a discussion in Appendix D.

## 4 EXPERIMENTAL RESULTS

In this section we systematically evaluate the aforementioned variants of local SGD on deep learning tasks. In summary, the main findings presented in Sections 4.1 and 4.2 below are:

*Local SGD,* on the one hand*, can serve as a communication-efficient alternative to mini-batch SGD for different practical purposes (communication restricted Scenario 1).* For example in Figure 2(a) (with fixed $B_{\mathrm{loc}}$ and $K$), in terms of $92.48\%$ best test accuracy achieved by our mini-batch SGD implementation for $K = 16$, practitioners can either choose to achieve reasonable good training quality ($91.2\%$, matching (He et al., 2016a)) with a $2.59\times$ speedup (time-to-accuracy) in training time ($H = 8$), or achieve slight better test performance ($92.57\%$) with slightly reduced communication efficiency ($1.76\times$ speedup for $H = 2$).

*Post-local SGD,* on the other hand, in addition to overcoming communication restrictions *does provide a state-of-the-art remedy for the generalization issue of large-batch training (Scenario 2).* Unlike local SGD with large $H$ and $K$ (e.g. $H = 16, K = 16$) which can encounter optimization issues during initial training (which can impact later generalization performance), we found that post-local SGD elegantly enables an ideal trade-off between optimization and generalization within a fixed

---

[2]The switching time $t'$ between the two phases in our experiments is determined by the first learning rate decay. However it could be tuned more generally aiming at capturing the time when trajectory starts to get into the influence basin of a local minimum (Robbins & Monro, 1985; Smith et al., 2018; Loshchilov & Hutter, 2017; Huang et al., 2017a). Results in Appendix B.4.2 and C.2 empirically evaluate the impact of different $t'$ on optimization and generalization.

training budget (e.g. Table 3, Table 5, Figure 3 and many others in Appendix C.5), and can achieve the same or even better performance than small mini-batch baselines.

**Setup.**  We briefly outline the general experimental setup, and refer to Appendix A for full details.

*Datasets.* We evaluate all methods on the following two main (standard) tasks: (1) Image classification for CIFAR-10/100 (Krizhevsky & Hinton, 2009), and (2) Image classification for ImageNet (Russakovsky et al., 2015). The detailed data augmentation scheme refers to Appendix A.

*Models.* We use ResNet-20 (He et al., 2016a) with CIFAR-10 as a base configuration to understand different properties of (post-)local SGD. We then empirical evaluate the large-batch training performance of post-local SGD, for ResNet-20, DensetNet-40-12 (Huang et al., 2017b) and WideResNet-28-10 (Zagoruyko & Komodakis, 2016) on CIFAR-10/100. Finally, we train ResNet-50 (He et al., 2016a) on ImageNet to investigate the accuracy and scalability of (post-)local SGD training.

*Implementation and platform.* Our algorithms are implemented[3] in PyTorch (Paszke et al., 2017), with a flexible configuration of the machine topology supported by Kubernetes. The cluster consists of Intel Xeon E5-2680 v3 servers and each server has 2 NVIDIA TITAN Xp GPUs. We use the notion $a \times b$-GPU to denote the topology of the cluster, i.e., $a$ nodes and each with $b$ GPUs.

*Specific learning schemes for large-batch SGD.* We rely on the recently proposed schemes for efficient large batch training (Goyal et al., 2017), which are formalized by (i) linearly scaling the learning rate w.r.t. the global mini-batch size; (ii) gradual warm-up of the learning rate from a small value. See Appendix A.3 for more details.

*Distributed training procedure on CIFAR-10/100.* The experiments follow the common mini-batch SGD training scheme for CIFAR (He et al., 2016a;b; Huang et al., 2017b) and all competing methods access the same total number of data samples (i.e. gradients) regardless of the number of local steps. Training ends when the distributed algorithms have accessed the same number of samples as the single-worker baseline. The data is disjointly partitioned and reshuffled globally every epoch. The learning rate scheme follows (He et al., 2016a; Huang et al., 2017b), where we drop the initial learning rate by a factor of 10 when the model has accessed 50% and 75% of the total number of training samples. Unless mentioned specifically, the used learning rate is scaled by the global mini-batch size ($BK$ for mini-batch SGD and $B_{\text{loc}}K$ for local SGD) where the initial learning rate is fine-tuned for each model and each task for the single worker. See Appendix A.4 for more details.

## 4.1 Superior Scalability of Local SGD over mini-batch SGD

First, we empirically study local SGD training for the communication restricted case (i.e. Scenario 1). As local SGD (with local batch size $B_{\text{loc}}$) needs $H$ times fewer communication rounds than mini-batch SGD (with the same batch size $B = B_{\text{loc}}$) we expect local SGD to significantly outperform mini-batch SGD in terms of time-to-accuracy and scalability and verify this experimentally. We further observe that local SGD shows better generalization performance than mini-batch SGD.

**Significantly better scalability when increasing the number of workers on CIFAR, in terms of time-to-accuracy.**  Figure 1 demonstrates the speedup in time-to-accuracy for training ResNet-20 for CIFAR-10, with varying number of GPUs $K$ and the number of local steps $H$, both from 1 to 16.

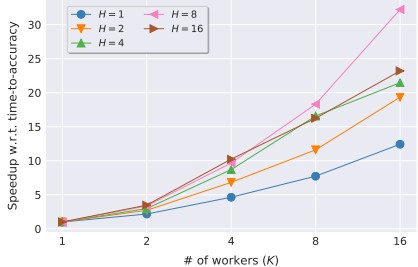

Table 1: Scaling behavior of **local SGD** in clock-time for increasing number of workers $K$, for different number of local steps $H$, for training **ResNet-20** on **CIFAR-10** with $B_{\text{loc}} = 128$. The reported **speedup** (averaged over three runs) is over single GPU training time for reaching the baseline top-1 test accuracy (91.2% as in (He et al., 2016a)). We use a $8 \times 2$-GPU cluster with 10 Gbps network. $H = 1$ recovers mini-batch SGD.

---

[3]Our code is available at `https://github.com/epfml/LocalSGD-Code`.

We demonstrate in Figure 1 that *local SGD scales 2× better than its mini-batch SGD counterpart, in terms of time-to-accuracy* as we increase the number of workers $K$ on a commodity cluster. The local update steps ($H$) result in a strong advantage over the standard large-batch training. Mini-batch SGD fixes the batch size to $B = B_{\text{loc}}$, and while increasing the number of workers $K$ gets impacted by the communication bottleneck (section 1), even as parallelism per device remains unchanged. In this experiment, local SGD on 8 GPUs even achieves a 2× lower time-to-accuracy than mini-batch SGD with 16 GPUs. Moreover, the (near) linear scaling performance for $H = 8$ in Figure 1, shows that the main hyper-parameter $H$ of local SGD is robust and consistently different from its mini-batch counterpart, when scaling the number of workers.

**Effectiveness and scalability of local SGD to even larger datasets (e.g., ImageNet) and larger clusters.** Local SGD presents a competitive alternative to the current large-batch ImageNet training methods. Figure 8 in Appendix B.3.2 shows that we can efficiently train state-of-the-art ResNet-50 (at least 1.5× speedup to reach 75% top-1 accuracy) for ImageNet (Goyal et al., 2017; You et al., 2017a) via local SGD on a $16 \times 2$-GPU cluster.

**Local SGD significantly outperforms mini-batch SGD at the same effective batch size and communication ratio.** Figure 2 compares local SGD with mini-batch SGD of the same effective batch size, that is the same number of gradient computations per communication round as described in Scenario 1 above (Section 1).

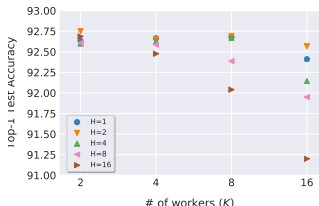 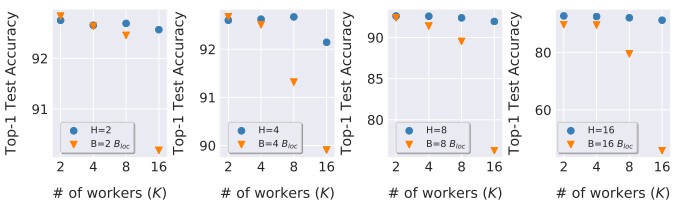

(a) Top-1 **test accuracy** of **local SGD**. $H = 1$ is mini-batch SGD with optimal hyper-parameters.

(b) Top-1 **test accuracy** of **local SGD** (circle) vs. **mini-batch SGD** (triangle), with same effective batch size $B = HB_{\text{loc}}$ per worker. The points for a given $K$ have the same communication cost.

Figure 2: Training **ResNet-20** on **CIFAR-10** under different $K$ and $H$, with fixed $B_{\text{loc}} = 128$. All results are averaged over three runs and all settings access to the same total number of training samples. We fine-tune the learning rate of mini-batch SGD for each setting.

## 4.2 (POST)-LOCAL SGD CLOSES THE GENERALIZATION GAP OF LARGE-BATCH TRAINING

Even though local SGD has demonstrated effective communication efficiency with guaranteed test performance (Scenario 1), we observed in the previous section that it still encounters difficulties when scaling to very large mini-batches (Figure 2(a)), though it is much less affected than mini-batch SGD (Figure 2(b)). Post-local SGD can address the generalization issue of large batch training (Scenario 2) as we show now. First, we would like to highlight some key findings in Table 2.

Table 2: Test performance (generalization) for local SGD variants and mini-batch SGD (highlighting selected data from Figure 2, Table 3 and Figure 3). Mini-batch SGD suffers from the generalization gap (and sometimes even from optimization issues due to insufficient training) when scaling to larger $H$ and $K$. Same experimental setup as in Figure 2 (fine-tuned mini-batch SGD baselines). The '→' denotes the transition at the first learning rate decay. The reported results are averaged over three runs.

| Algorithm | Top-1 acc. | Effect. batch size | Algorithm | Top-1 acc. | Effect. batch size |
|---|---|---|---|---|---|
| Mini-batch SGD ($K = 4$) | 92.6% | $KB = 512$ | Mini-batch SGD ($K = 16$) | 92.5% | $KB = 2048$ |
| Mini-batch SGD ($K = 4$) | 89.5% | $KB = 8192$ | Mini-batch SGD ($K = 16$) | 76.3% | $KB = 16384$ |
| Mini-batch SGD ($K = 4$) | 92.5% | $KB = 512 \rightarrow 8192$ | Mini-batch SGD ($K = 16$) | 92.0% | $KHB_{\text{loc}} = 2048 \rightarrow 16384$ |
| Local SGD ($H = 16, K = 4$) | 92.5% | $KHB_{\text{loc}} = 8192$ | Local SGD ($H = 8, K = 16$) | 92.0% | $KHB_{\text{loc}} = 16384$ |
| Post-local SGD ($H = 16, K = 4$) | **92.7%** | $KHB_{\text{loc}} = 512 \rightarrow 8192$ | Post-local SGD ($H = 8, K = 16$) | **92.9%** | $KHB_{\text{loc}} = 2048 \rightarrow 16384$ |

We now focus on the generalization issues (Section 1) of large-batch SGD, isolating the potential negative impact from the optimization difficulty (e.g. the insufficient training epochs (Shallue et al., 2018)) from the performance on the test set. Our evaluation starts from a (standard) constant effective mini-batch size of 2048 or 4096 for mini-batch SGD[4] on CIFAR dataset, as in (Keskar et al., 2017; Hoffer et al., 2017; Yao et al., 2018).

**Post-local SGD generalizes better and faster than mini-batch SGD.** Table 3 summarizes the generalization performance of post-local SGD on large batch size ($K = 16, B_{loc} = 128$) across different architectures on CIFAR tasks for $H = 16$ and $H = 32$. Under the same setup, Table 9 in the Appendix C.3 evaluates the speedup of training, while Figure 1 demonstrates the learning curves of mini-batch SGD and post-local SGD, highlighting the generalization difficulty of large-batch SGD. We can witness that post-local SGD achieves at least $1.3\times$ speedup over the whole training procedure compared to the mini-batch SGD counterpart ($K = 16, B = 128$), while enjoying the significantly improved generalization performance.

Table 3: Top-1 **test accuracy** of training different CNN models via **post-local SGD** on $K = 16$ GPUs with a large global batch size ($KB_{loc} = 2048$). The reported results are the average of three runs and all settings access to the same total number of training samples. We compare to small and large mini-batch baselines. The $\star$ indicates a fine-tuned learning rate, where the tuning procedure refers to Appendix A.4.

| | CIFAR-10 | | | | CIFAR-100 | | | |
|---|---|---|---|---|---|---|---|---|
| | Small batch baseline $\star$ | Large batch baseline $\star$ | Post-local SGD (H=16) | Post-local SGD (H=32) | Small batch baseline $\star$ | Large batch baseline $\star$ | Post-local SGD (H=16) | Post-local SGD (H=32) |
| | $K=2, B=128$ | $K=16, B=128$ | $K=16, B_{loc}=128$ | $K=16, B_{loc}=128$ | $K=2, B=128$ | $K=16, B=128$ | $K=16, B_{loc}=128$ | $K=16, B_{loc}=128$ |
| **ResNet-20** | 92.63 $\pm 0.26$ | 92.48 $\pm 0.17$ | 92.80 $\pm 0.16$ | **93.02** $\pm 0.24$ | 68.84 $\pm 0.06$ | 68.17 $\pm 0.18$ | 69.24 $\pm 0.26$ | **69.38** $\pm 0.20$ |
| **DenseNet-40-12** | 94.41 $\pm 0.14$ | 94.36 $\pm 0.20$ | 94.43 $\pm 0.12$ | **94.58** $\pm 0.11$ | 74.85 $\pm 0.14$ | 74.08 $\pm 0.46$ | 74.45 $\pm 0.30$ | **75.03** $\pm 0.05$ |
| **WideResNet-28-10** | 95.89 $\pm 0.10$ | 95.43 $\pm 0.37$ | **95.94** $\pm 0.06$ | 95.76 $\pm 0.25$ | 79.78 $\pm 0.16$ | 79.31 $\pm 0.23$ | 80.28 $\pm 0.13$ | **80.65** $\pm 0.16$ |

We further demonstrate the generalization performance and the scalability of post-local SGD, for diverse tasks (e.g., Language Modeling), and for even larger global batch sizes ($KB_{loc} = 4096$ for CIFAR-100 and 4096 and 8192 respectively for ImageNet), in Appendix C.5 . For example, Table 11 presents the severe generalization issue (2% drop) of the fine-tuned large-batch SGD training ($KB = 4096$) for above three CNNs on CIFAR-100, and cannot be addressed by increasing the training steps (Table 12). Post-local SGD ($KB_{loc} = 4096$) with default hyper-parameters can perfectly close this generalization gap or even better than the fine-tuned small mini-batch baselines[5]. For ImageNet training in Figure 16, the post-local SGD outperforms mini-batch SGD baseline for both of $KB = 4096$ (76.18 and 75.87 respectively) and $KB = 8192$ (75.65 and 75.64 respectively) with $1.35\times$ speedup for the post-local SGD training phase.

**The effectiveness of post-local SGD training for different $H$ and $K$.** As seen in Figure 3(a), applying any number of local steps over the case of large-batch training (when $KB_{loc} = 2048$) improves the generalization performance compared to mini-batch SGD. Figure 3(b) illustrates that post-local SGD is better than mini-batch SGD in general for different number of workers $K$ (as well as different $KB_{loc}$). Thus, post-local SGD presents consistently excellent generalization performance.

**Post-local SGD can improve the training efficiency upon other compression techniques.** The results in Table 4 illustrate that post-local SGD can be combined with other communication-efficient techniques (e.g. the sign-based compression schemes) for further improved communication efficiency and better test generalization performance.

**Post-local SGD can improve upon other SOTA optimizers.** The benefits of post-local SGD training on ImageNet are even more pronounced for larger batches (e.g. $KB_{loc} > 8192$) (Goyal et al.,

---

[4]It is less challenging (e.g. comparing the first and third rows in the right column of Table 2) to train with mini-batch SGD for medium-size, constant effective mini-batch size. In our evaluation, the training of the post-local SGD and mini-batch SGD will start from the same effective mini-batch size with the same number of workers $K$. Even under this relaxed comparison, post-local SGD still significantly outperform mini-batch SGD.

[5]We omit the comparison with other noise injection methods as none of them (Neelakantan et al., 2015; Wen et al., 2019) can completely address the generalization issue even for CIFAR dataset.

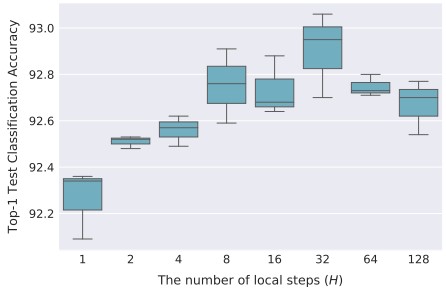
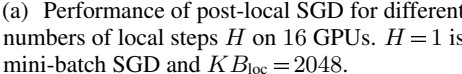

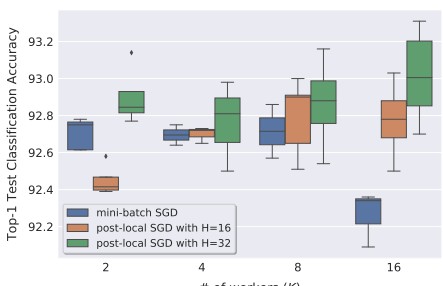

(a) Performance of post-local SGD for different numbers of local steps $H$ on 16 GPUs. $H = 1$ is mini-batch SGD and $KB_{\text{loc}} = 2048$.

(b) Performance of post-local SGD for different numbers of workers $K$ for $H = 16$ and $H = 32$. The local batch size is fixed to $B_{\text{loc}} = B = 128$ both for local and mini-batch SGD).

Figure 3: Top-1 **test accuracy** of training **ResNet-20** on **CIFAR-10**. Box-plot figures are derived from 3 runs.

Table 4: Top-1 **test accuracy** of training **ResNet-20** on **CIFAR** via sign-based compression scheme and **post-local SGD** ($KB_{\text{loc}} = 2048$ and $K = 16$). $H = 1$ in the table for the simplicity corresponds to the original sign-based algorithm and we fine-tune their hyper-parameters. The reported results are the average of three runs with different seeds. The learning rate schedule is fixed for different local update steps (for $H > 1$). For the detailed tuning procedure, training scheme, as well as the pseudo code of the used variants, we refer to Appendix C.5.5.

| | CIFAR-10 | | | | CIFAR-100 | | | |
|---|---|---|---|---|---|---|---|---|
| | $H = 1$ | $H = 16$ | $H = 32$ | $H = 64$ | $H = 1$ | $H = 16$ | $H = 32$ | $H = 64$ |
| **signSGD** (Bernstein et al., 2018) | 91.61 $\pm 0.28$ | 91.77 $\pm 0.04$ | **92.22** $\pm 0.11$ | 92.00 $\pm 0.15$ | 67.21 $\pm 0.11$ | 67.43 $\pm 0.36$ | **68.32** $\pm 0.18$ | 68.12 $\pm 0.11$ |
| **EF-signSGD** (Karimireddy et al., 2019) | 92.45 $\pm 0.28$ | 92.63 $\pm 0.16$ | 92.72 $\pm 0.05$ | **92.76** $\pm 0.06$ | 68.19 $\pm 0.10$ | 69.14 $\pm 0.36$ | 69.00 $\pm 0.14$ | **69.49** $\pm 0.21$ |

2017; Shallue et al., 2018; Golmant et al., 2018). Considering other optimizers, Table 5 below shows that post-local SGD can improve upon LARS[6] (You et al., 2017a).

Table 5: The Top-1 test accuracy of training **ResNet50** on **ImageNet**. We report the performance (w/ or w/o post-local SGD) achieved by using SGD with Nesterov Momentum, learning schemes in (Goyal et al., 2017), and LARS. We follow the general experimental setup described in Section A.4.2, and use $K = 32$ and $H = 4$.

| | SGD + Momentum + LARS | SGD + Momentum + LARS + Post-local SGD |
|---|---|---|
| $KB_{\text{loc}} = 8,192$ | 75.99 | 76.13 |
| $KB_{\text{loc}} = 16,384$ | 74.52 | 75.23 |

## 5 DISCUSSION AND INTERPRETATION

Generalization issues of large-batch SGD training are not yet very well understood from a theoretical perspective. Hence, a profound theoretical study of the generalization local SGD is beyond the scope of this work but we hope the favorable experimental results will trigger future research in this direction. In this section we discuss some related work and we argue that local SGD can be seen as a way to inject and control stochastic noise to the whole training procedure.

**Connecting Local Updates with Stochastic Noise Injection.** The update eq. (1) can alternatively be written as

$$\boldsymbol{w}_{t+1} = \boldsymbol{w}_t - \gamma \hat{\boldsymbol{g}}_t = \boldsymbol{w}_t - \gamma \boldsymbol{g}_t + \gamma \big( \boldsymbol{g}_t - \hat{\boldsymbol{g}}_t \big) = \boldsymbol{w}_t - \gamma \boldsymbol{g}_t + \gamma \boldsymbol{\epsilon}, \tag{3}$$

where $\hat{\boldsymbol{g}}_t := \frac{1}{B} \sum_i \nabla f_i(\boldsymbol{w}_t)$, $\boldsymbol{g}_t := \mathbb{E} \hat{\boldsymbol{g}}_t = \nabla f(\boldsymbol{w}_t)$ and $\boldsymbol{\epsilon} := \boldsymbol{g}_t - \hat{\boldsymbol{g}}_t$. For most data-sets the noise $\boldsymbol{\epsilon}$ can be well approximated by a zero-mean Gaussian with variance $\boldsymbol{\Sigma}(\boldsymbol{w})$. The variance matrix $\boldsymbol{\Sigma}(\boldsymbol{w})$, following Hoffer et al. (2017) and Hu et al. (2017), for uniform sampling of mini-batch indices can be approximated as

$$\boldsymbol{\Sigma}(\boldsymbol{w}) \approx \left( \tfrac{1}{B} - \tfrac{1}{N} \right) \boldsymbol{K}(\boldsymbol{w}) = \left( \tfrac{1}{B} - \tfrac{1}{N} \right) \left( \tfrac{1}{N} \sum_{i=1}^{N} \nabla f_i(\boldsymbol{w}) \nabla f_i(\boldsymbol{w})^{\top} \right). \tag{4}$$

---

[6]The LARS implementation follows the code github.com/NVIDIA/apex for mixed precision and distributed training in Pytorch. LARS uses layer-wise learning rate based on the ratio between $\|\boldsymbol{w}\|_2$ and $\|\nabla f_i(\boldsymbol{w})\|_2$, thus our post-local SGD can be simply integrated without extra modification and parameter synchronization.

Recent works (Jastrzębski et al., 2018; Li et al., 2017; Mandt et al., 2017; Zhu et al., 2019) interpret eq. (3) as an Euler-Maruyama approximation to the continuous-time Stochastic Differential Equation (SDE): $d\boldsymbol{w}_t = -\boldsymbol{g}_t dt + \sqrt{\gamma \frac{N-B}{BN}} \boldsymbol{R}(\boldsymbol{w}) d\boldsymbol{\theta}(t)$ where $\boldsymbol{R}(\boldsymbol{w})\boldsymbol{R}(\boldsymbol{w})^\top = \boldsymbol{K}(\boldsymbol{w})$ and $\boldsymbol{\theta}(t) \sim \mathcal{N}(0, \boldsymbol{I})$. When $B \ll N$ (in eq. (4)), the learning rate and the batch size only appear in the SDE through the ratio $\rho := \gamma B^{-1}$. Jastrzębski et al. (2018) argue that thus mainly the ratio $\rho$ controls the stochastic noise and training dynamics, and that $\rho$ positively correlates with wider minima and better generalization—matching with recent experimental successes of ImageNet training (Goyal et al., 2017).

This explanation fails in the large batch regime. When the batch size grows too large, the relative magnitude of the stochastic noise decreases as $\gamma B^{-1} \not\approx \gamma(N-B)/BN$, i.e. $\rho$ is not anymore uniquely describing the training dynamics for large batch size and small dataset size when $B \not\ll N$ (Jastrzębski et al., 2018). This could be a reason why the generalization difficulty of large-batch training remains, as clearly illustrated e.g. in Figure 1, Table 3, Figure 3(b), and as well as in (Shallue et al., 2018; Golmant et al., 2018).

The local update step of local SGD is a natural and computation-free way to inject well-structured stochastic noise to the SGD training dynamics. By using the same ratio $\frac{\gamma}{B}$ as mini-batch SGD, the main difference of the local SGD training dynamics comes from the stochastic noise $\boldsymbol{\epsilon}$ during the local update phase ($K$ times smaller local mini-batch with $H$ local update steps). The $\boldsymbol{\epsilon}$ will be approximately sampled with the variance matrix $K\boldsymbol{\Sigma}(\boldsymbol{w})$ instead of $\boldsymbol{\Sigma}(\boldsymbol{w})$ in mini-batch SGD, causing the stochastic noise determined by $K$ and $H$ to increase. This could be one of the reasons for post-local SGD to generalize as good as mini-batch SGD (small mini-batch size) and leading to flatter minima than large-batch SGD in our experiments. Similar positive effects of adding well-structured stochastic noise to SGD dynamics on non-convex problems have recently also been observed in (Smith & Le, 2018; Chaudhari & Soatto, 2018; Zhu et al., 2019; Xing et al., 2018; Wen et al., 2019).

## 5.1 POST-LOCAL SGD CONVERGES TO FLATTER MINIMA

Figure 4(a) evaluates the spectrum of the Hessian for different local minima. We observe that large-batch SGD tends to get stuck at locations with high Hessian spectrum while post-local SGD tends to prefer low curvature solutions with better generalization error. Figure 4(b) linearly interpolates two minima obtained by mini-batch SGD and post-local SGD, and Figure 13 (in the Appendix) visualizes the sharpness of the model trained by different methods. These results give evidence that post-local SGD converges to flatter minima than mini-batch SGD for training ResNet-20 on CIFAR-10.

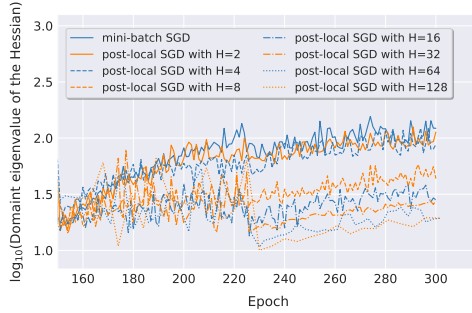

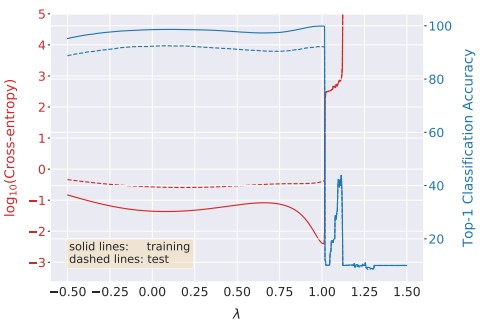

(a) The dominant eigenvalue of the Hessian, for model trained from different schemes. It is evaluated on the test dataset per epoch, and only the phase related to the post-local SGD strategy is visualized. Mini-batch SGD or post-local SGD with small $H$ (e.g., $H = 2, 4$) have noticeably larger dominant eigenvalue.

(b) 1-d linear interpolation between models $\boldsymbol{w}_{\text{post-local SGD}}$ ($\lambda = 0$) and $\boldsymbol{w}_{\text{mini-batch SGD}}$ ($\lambda = 1$), i.e., $\hat{\boldsymbol{w}} = \lambda \boldsymbol{w}_{\text{mini-batch SGD}} + (1 - \lambda) \boldsymbol{w}_{\text{post-local SGD}}$. The solid lines correspond to evaluate $\hat{\boldsymbol{w}}$ on the whole training dataset while the dashed lines are on the test dataset. The model parameters only differ from the post-local SGD phase.

Figure 4: Understanding the generalization ability of post-local SGD for large-batch training (**ResNet-20** on **CIFAR-10** with $BK = B_{\text{loc}}K = 2048$). We use fixed $B = B_{\text{loc}} = 128$ with $K = 16$ GPUs. The detailed experimental setup as well as more visualization of results are available in Appendix C.4.

## 6 CONCLUSION

We leverage the idea of local SGD for training in distributed and heterogeneous environments. Ours is the first work to extensively study the trade-off between communication efficiency and generalization performance of local SGD. Our local SGD variant, called post-local SGD, not only outperforms large-batch SGD's generalization performance but also matches that of small-batch SGD. In our experiments post-local SGD converged to flatter minima compared to traditional large-batch SGD, which partially explains its improved generalization performance. We also provide extensive experiments with another variant hierarchical local SGD, showing its adaptivity to available system resources. Overall, local SGD comes off as a simpler and more efficient algorithm, replacing complex ad-hoc tricks used for current large-batch SGD training.

## ACKNOWLEDGEMENTS

The authors thank the anonymous reviewers and Thijs Vogels for their precious comments and feedback. We acknowledge funding from SNSF grant 200021_175796, as well as a Google Focused Research Award.

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

# Supplementary Material

For easier navigation through the paper and the extensive appendix, we include a table of contents.

CONTENTS

## A   Details on Deep Learning Experimental Setup

### A.1   Dataset

We use the following tasks.

- Image classification for CIFAR-10/100 (Krizhevsky & Hinton, 2009). Each consists of a training set of 50K and a test set of 10K color images of $32 \times 32$ pixels, as well as 10 and 100 target classes respectively. We adopt the standard data augmentation scheme and preprocessing scheme (He et al., 2016a; Huang et al., 2016). For preprocessing, we normalize the data using the channel means and standard deviations.
- Image classification for ImageNet (Russakovsky et al., 2015). The ILSVRC 2012 classification dataset consists of 1.28 million images for training, and 50K for validation, with 1K target classes. We use ImageNet-1k (Deng et al., 2009) and adopt the same data preprocessing and augmentation scheme as in He et al. (2016a;b); Simonyan & Zisserman (2015). The network input image is a $224 \times 224$ pixel random crop from augmented images, with per-pixel mean subtracted.
- Language Modeling for WikiText-2 (Merity et al., 2017). WikiText-2 is sourced from curated Wikipedia articles. It is frequently used for machine translation and language modelling, and features a vocabulary of over $30,000$ words. Compared to the preprocessed version of Penn Treebank (PTB), WikiText-2 is over 2 times larger.

### A.2   Models and Model Initialization

We use ResNet-20 (He et al., 2016a) with CIFAR-10 as a base configuration to understand different properties of (post-)local SGD. We then empirically evaluate the large-batch training performance of post-local SGD, for ResNet-20, DensetNet-40-12 (Huang et al., 2017b) and WideResNet-28-10 (Zagoruyko & Komodakis, 2016) on CIFAR-10/100, and for LSTM on WikiText-2 (Merity et al., 2017). Finally, we train ResNet-50 (He et al., 2016a) on ImageNet to investigate the accuracy and scalability of (post-)local SGD training.

For the weight initialization we follow Goyal et al. (2017), where we adopt the initialization introduced by He et al. (2015) for convolutional layers and initialize fully-connected layers by a zero-mean Gaussian distribution with the standard deviation of 0.01.

Table 6 demonstrates the scaling ratio of our mainly used Neural Network architectures. The scaling ratio (You et al., 2017b) identifies the ratio between computation and communication, wherein

DNN models, the computation is proportional to the number of floating point operations required for processing an input while the communication is proportional to model size (or the number of parameters). Our local SGD training scheme will show more advantages over models with small "computation and communication scaling ratio".

Table 6: Scaling ratio for different models.

| Model | Communication # parameters | Computation # flops per image | Computation/Communication scaling ratio |
|---|---|---|---|
| ResNet-20 (CIFAR-10) | 0.27 million | 0.041 billion | 151.85 |
| ResNet-20 (CIFAR-100) | 0.27 million | 0.041 billion | 151.85 |
| ResNet-50 (ImageNet-1k) | 25.00 million | 7.7 billion | 308.00 |
| DenseNet-40-12 (CIFAR-10) | 1.06 million | 0.28 billion | 264.15 |
| DenseNet-40-12 (CIFAR-100) | 1.10 million | 0.28 billion | 254.55 |
| WideResNet-28-10 (CIFAR-10) | 36.48 million | 5.24 billion | 143.64 |
| WideResNet-28-10 (CIFAR-100) | 36.54 million | 5.24 billion | 143.40 |

### A.3 Large Batch Learning Schemes

The work of Goyal et al. (2017) proposes common configurations to tackle large-batch training for the ImageNet dataset. We specifically refer to their crucial techniques w.r.t. learning rate as "large batch learning schemes" in our main text. For a precise definition, this is formalized by the following two configurations:

- **Scaling the learning rate**: When the mini-batch size is multiplied by $k$, multiply the learning rate by $k$.
- **Learning rate gradual warm-up**: We gradually ramp up the learning rate from a small to a large value. In (our) experiments, with a large mini-batch of size $kn$, we start from a learning rate of $\eta$ and increment it by a constant amount at each iteration such that it reaches $\hat{\eta} = k\eta$ after 5 epochs. More precisely, the incremental step size for each iteration is calculated from $\frac{\hat{\eta}-\eta}{5N/(kn)}$, where $N$ is the number of total training samples, $k$ is the number of computing units and $n$ is the local mini-batch size.

### A.4 Hyperparameter Choices and Training Procedure, over Different Models/Datasets

#### A.4.1 CIFAR-10/CIFAR-100

The experiments follow the common mini-batch SGD training scheme for CIFAR (He et al., 2016a;b; Huang et al., 2017b) and all competing methods access the same total amount of data samples regardless of the number of local steps. The training procedure is terminated when the distributed algorithms have accessed the same number of samples as a standalone worker would access. For example, ResNet-20, DensetNet-40-12 and WideResNet-28-10 would access 300, 300 and 250 epochs respectively. The data is partitioned among the GPUs and reshuffled globally every epoch. The local mini-batches are then sampled among the local data available on each GPU, and its size is fixed to $B_{\text{loc}} = 128$.

The learning rate scheme follows works (He et al., 2016a; Huang et al., 2017b), where we drop the initial learning rate by 10 when the model has accessed 50% and 75% of the total number of training samples. The initial learning rates of ResNet-20, DensetNet-40-12 and WideResNet-28-10 are fine-tuned on single GPU (which are 0.2, 0.2 and 0.1 respectively), and can be scaled by the global mini-batch size when using large-batch learning schemes.

In addition to this, we use a Nesterov momentum of 0.9 without dampening, which is applied independently to each local model. For all architectures, following He et al. (2016a), we do not

apply weight decay on the learnable Batch Normalization (BN) coefficients. The weight decay of ResNet-20, DensetNet-40-12 and WideResNet-28-10 are $1e$-$4$, $1e$-$4$ and $5e$-$4$ respectively. For the BN for distributed training we again follow Goyal et al. (2017) and compute the BN statistics independently for each worker.

Unless mentioned specifically, local SGD uses the exact same optimization scheme as mini-batch SGD.

**The procedure of fine-tuning.** There is no optimal learning rate scaling rule for large-batch SGD across different mini-batch sizes, tasks and architectures, as revealed in the Figure 8 of Shallue et al. (2018). Our tuning procedure is built on the insight of their Figure 8, where we grid-search the optimal learning rate for each mini-batch size, starting from the linearly scaled learning rate. For example, compared to mini-batch size 128, mini-batch size 2048 with default large-batch learning schemes need to linearly scale the learning rate by the factor of 16. In order to find out its optimal learning rate, we will evaluate a linear-spaced grid of five different factors (i.e., $\{15, 15.5, 16, 16.5, 17\}$). If the best performance was ever at one of the extremes of the grid, we would try new grid points so that the best performance was contained in the middle of the parameters. Please note that this unbounded grid search—even though initialized at the linearly scaled learning rate—does also cover other suggested scaling heuristics, for instance square root scaling, and finds the best scaling for each scenario.

Note that in our experiments of large-batch SGD, either with the default large-batch learning schemes, or tuning/using the optimal learning rate, we always warm-up the learning rate for the first 5 epochs.

### A.4.2 IMAGENET

ResNet-50 training is limited to 90 passes over the data in total, and the data is disjointly partitioned and is re-shuffled globally every epoch. All competing methods access the same total number of data samples (i.e. gradients) regardless of the number of local steps. We adopt the large-batch learning schemes as in Goyal et al. (2017) below. We linearly scale the learning rate based on $\left(\text{Number of GPUs} \times \frac{0.1}{256} \times B_{\text{glob}}\right)$ where 0.1 and 256 is the base learning rate and mini-batch size respectively for standard single GPU training. The local mini-batch size is set to 128. For learning rate scaling, we perform gradual warmup for the first 5 epochs, and decay the scaled learning rate by the factor of 10 when local models have access $30, 60, 80$ epochs of training samples respectively.

### A.5 SYSTEM PERFORMANCE EVALUATION

Figure 5 investigates the increased latency of transmitting data among CPU cores.

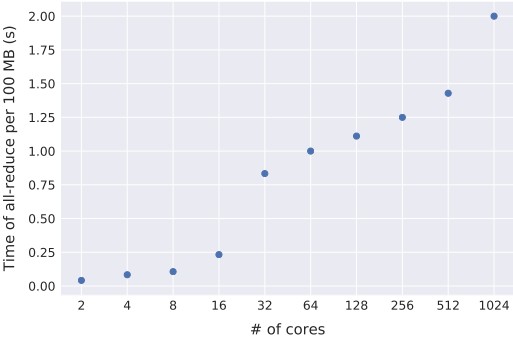

Figure 5: The data transmission cost (in seconds) of an all-reduce operation for 100 MB, over the different number of cores, using PyTorch's built-in MPI all-reduce operation. Each evaluation is the average result of 100 data transmissions on a Kubernetes cluster. The network bandwidth is 10 Gbps, and we use 48 cores per physical machine.

Table 7 evaluates the time of running forward and backward with different mini-batch size, for training ResNet20 on CIFAR-10. We can witness that a larger mini-batch size we use, the better parallelism can a GPU have.

Table 7: The system performance of running forward and backward pass on a single GPU, for training ResNet20 on CIFAR-10. The "Ratio" indicates the $\frac{\text{Time of evaluating 4096 samples with specified mini-batch size}}{\text{Time of evaluating 4096 samples with mini-batch size 4096}}$. The "Time" is in seconds.

| | Titan XP | | Tesla V100 | |
|---|---|---|---|---|
| Mini-Batch Size | Time per iteration (over 100 iterations) | Ratio | Time per iteration (over 100 iterations) | Ratio |
| 32 | 0.058 | 1.490 | 0.028 | 9.028 |
| 64 | 0.100 | 1.284 | 0.030 | 4.836 |
| 128 | 0.175 | 1.124 | 0.034 | 2.741 |
| 256 | 0.323 | 1.037 | 0.043 | 1.733 |
| 512 | 0.737 | 1.183 | 0.073 | 1.471 |
| 1024 | 1.469 | 1.179 | 0.124 | 1.249 |
| 2048 | 2.698 | 1.083 | 0.212 | 1.068 |
| 4096 | 4.983 | 1 | 0.397 | 1 |

# B    LOCAL SGD TRAINING

## B.1    FORMAL DEFINITION OF THE LOCAL SGD ALGORITHM

---

**Algorithm 1** *Local SGD*

---

**input:** the initial model $\boldsymbol{w}_{(0)}$;
**input:** training data with labels $\mathcal{I}$;
**input:** mini-batch of size $B_{\text{loc}}$ per local model;
**input:** step size $\eta$, and momentum $m$ (optional);
**input:** number of synchronization steps $T$;
**input:** number of local steps $H$;
**input:** number of nodes $K$.
1: synchronize to have the same initial models $\boldsymbol{w}_{(0)}^k := \boldsymbol{w}_{(0)}$.
2: **for all** $k := 1, \ldots, K$ **do in parallel**
3:     **for** $t := 1, \ldots, T$ **do**
4:         **for** $h := 1, \ldots, H$ **do**
5:             sample a mini-batch from $\mathcal{I}_{(t)+h-1}^k$.
6:             compute the gradient

$$\boldsymbol{g}_{(t)+h-1}^k := \tfrac{1}{B_{\text{loc}}} \textstyle\sum_{i \in \mathcal{I}_{(t)+h-1}^k} \nabla f_i\!\left(\boldsymbol{w}_{(t)+h-1}^k\right).$$

7:             update the local model to

$$\boldsymbol{w}_{(t)+h}^k := \boldsymbol{w}_{(t)+h-1} - \gamma_{(t)} \boldsymbol{g}_{(t)+h-1}^k \, .$$

8:         **end for**
9:         all-reduce aggregation of the gradients

$$\Delta_{(t)}^k := \boldsymbol{w}_{(t)}^k - \boldsymbol{w}_{(t)+H}^k \, .$$

10:         get new global (synchronized) model $\boldsymbol{w}_{(t+1)}^k$ for all $K$ nodes:

$$\boldsymbol{w}_{(t+1)}^k := \boldsymbol{w}_{(t)}^k - \gamma_{(t)} \tfrac{1}{K} \textstyle\sum_{i=1}^K \Delta_{(t)}^k$$

11:     **end for**
12: **end for**

---

## B.2    NUMERICAL ILLUSTRATION OF LOCAL SGD ON A CONVEX PROBLEM

In addition to our deep learning experiments, we first illustrate the convergence properties of local SGD on a small scale convex problem. For this, we consider logistic regression on the `w8a` dataset[7]

---
[7] www.csie.ntu.edu.tw/~cjlin/libsvmtools/datasets/binary.html

$(d = 300, n = 49749)$. We measure the number of iterations to reach the target accuracy $\epsilon = 0.005$. For each combination of $H$, $B_{\text{loc}}$ and $K$ we determine the best learning rate by extensive grid search (cf. paragraph below for the detailed experimental setup). In order to mitigate extraneous effects on the measured results, we here measure time in discrete units, that is we count the number of stochastic gradient computations and communication rounds, and assume that communication of the weights is $25\times$ more expensive than a gradient computation, for ease of illustration.

Figure 6(a) shows that different combinations of the parameters $(B_{\text{loc}}, H)$ can impact the convergence time for $K = 16$. Here, local SGD with $(16, 16)$ converges more than $2\times$ faster than for $(64, 1)$ and $3\times$ faster than for $(256, 1)$.

Figure 6(b) depicts the speedup when increasing the number of workers $K$. Local SGD shows the best speedup for $H = 16$ on a small number of workers, while the advantage gradually diminishes for very large $K$.

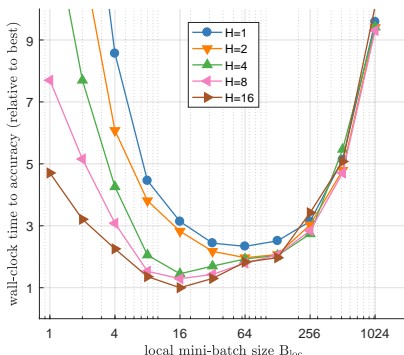

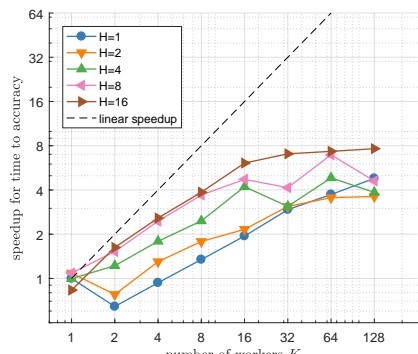

(a) **Time** (relative to best method) to solve a regularized logistic regression problem to target accuracy $\epsilon = 0.005$ for $K = 16$ workers for $H \in 1, 2, 4, 8, 16$ and local mini-batch size $B_{\text{loc}}$. We simulate the network traffic under the assumption that communication is $25\times$ slower than a stochastic gradient computation.

(b) **Speedup** over the number of workers $K$ to solve a regularized logistic regression problem to target accuracy $\epsilon = 0.005$, for $B_{\text{loc}} = 16$ and $H \in 1, 2, 4, 8, 16$. We simulate the network traffic under the assumption that communication is $25\times$ slower than a stochastic gradient computation.

Figure 6: Numerical illustration of local SGD on a convex problem.

**Experimental Setup for Convex Experiments** For the illustrative experiments here we study the convergence of local SGD on the logistic regression problem, $f(\boldsymbol{w}) = \frac{1}{n} \sum_{i=1}^{n} \log(1 + \exp(-b_i \mathbf{a}_i^\top \boldsymbol{w})) + \frac{\lambda}{2}\|\boldsymbol{w}\|^2$, where $\mathbf{a}_i \in \mathbb{R}^d$ and $b_i \in \{-1, +1\}$ are the data samples, and regularization parameter $\lambda = 1/n$. For each run, we initialize $\boldsymbol{w}_0 = \mathbf{0}_d$ and measure the number of stochastic gradient evaluations (and communication rounds) until be best of last iterate and weighted average of the iterates reaches the target accuracy $f(\boldsymbol{w}_t) - f^\star \leqslant \epsilon := 0.005$, with $f^\star := 0.126433176216545$. For each configuration $(K, H, B_{\text{loc}})$, we report the best result found with any of the following two stepsizes: $\gamma_t := \min(32, \frac{cn}{t+1})$ and $\gamma_t = 32c$. Here $c$ is a parameter that can take the values $c = 2^i$ for $i \in \mathbb{Z}$. For each stepsize we determine the best parameter $c$ by a grid search, and consider parameter $c$ optimal, if parameters $\{2^{-2}c, 2^{-1}c, 2c, 2^2c\}$ yield worse results (i.e. more iterations to reach the target accuracy).

### B.3 MORE RESULTS ON LOCAL SGD TRAINING

#### B.3.1 TRAINING CIFAR-10 VIA LOCAL SGD

**Better communication efficiency, with guaranteed test accuracy.** Figure 7 shows that *local SGD is significantly more communication efficient while guaranteeing the same accuracy and enjoys faster convergence speed.* In Figure 7, the local models use a fixed local mini-batch size $B_{\text{loc}} = 128$ for all updates. All methods run for the same number of total gradient computations. Mini-batch

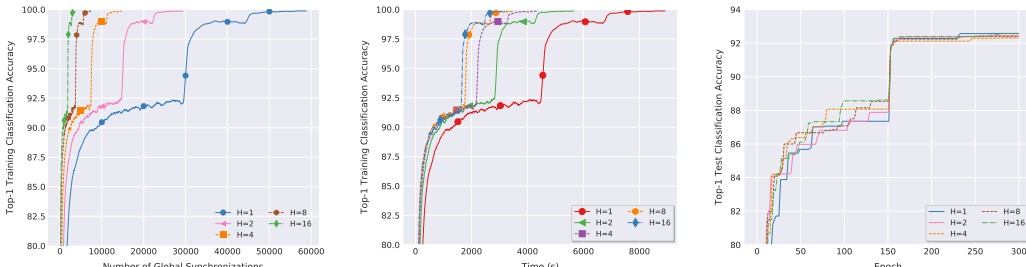

(a) Training accuracy vs. number of global synchronization rounds.

(b) Training accuracy vs. training time.

(c) Test accuracy vs. number of epochs.

Figure 7: Training **CIFAR-10** with **ResNet-20** via **local SGD** ($2 \times 1$-GPU). The local batch size $B_{loc}$ is fixed to 128, and the number of local steps $H$ is varied from 1 to 16. All the experiments are under the same training configurations.

SGD—the baseline method for comparison—is a special case of local SGD with $H = 1$, with full global model synchronization for each local update. We see that local SGD with $H > 1$, as illustrated in Figure 7(a), by design does $H$ times less global model synchronizations, alleviating the communication bottleneck while accessing the same number of samples (see section 1). The impact of local SGD training upon the total training time is more significant for larger number of local steps $H$ (i.e., Figure 7(b)), resulting in an at least $3\times$ speed-up when comparing mini-batch $H = 1$ to local SGD with $H = 16$. The final training accuracy remains stable across different $H$ values, and there is no difference or negligible difference in test accuracy (Figure 7(c)).

### B.3.2 TRAINING IMAGENET VIA LOCAL SGD

Figure 8 shows the effectiveness of scaling local SGD to the challenging ImageNet dataset. We limit ResNet-50 training to 90 passes over the data in total, and use the standard training configurations as mentioned in Appendix A.4.2.

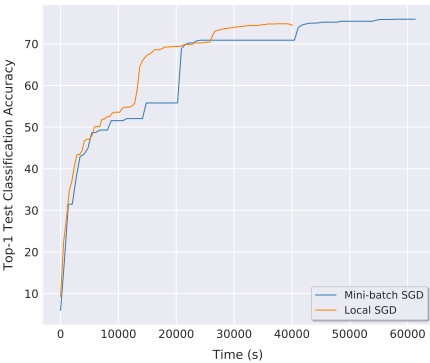

Figure 8: The performance of **local SGD** trained on **ImageNet-1k** with **ResNet-50**. We evaluate the model performance on test dataset after each complete accessing of the whole training samples. We apply the large-batch learning schemes (Goyal et al., 2017) to the ImageNet for these two methods. For local SGD, the number of local steps is set to $H = 8$.

Moreover, in our ImageNet experiment, the initial phase of local SGD training follows the theoretical assumption mentioned in Subsection 2, and thus we gradually warm up the number of local steps from 1 to the desired value $H$ during the first few epochs of the training. We found that exponentially increasing the number of local steps from 1 by the factor of 2 (until reaching the expected number of local steps) performs well. For example, our ImageNet training uses $H = 8$, so the number of local steps for the first three epochs is $1, 2, 4$ respectively.

### B.3.3 LOCAL SGD SCALES TO LARGER BATCH SIZES THAN MINI-BATCH SGD

The empirical studies (Shallue et al., 2018; Golmant et al., 2018) reveal the regime of maximal data parallelism across different tasks and models, where the large-batch training would reach the limit and additional parallelism provides no benefit whatsoever.

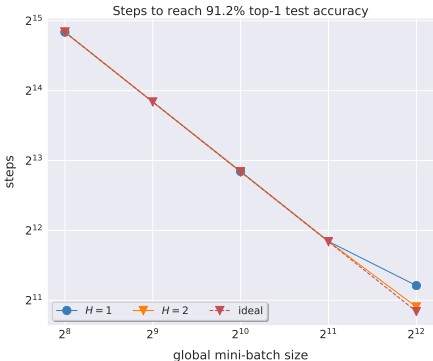

Figure 9: The relationship between steps to top-1 **test accuracy** and batch size, of training **ResNet-20** on **CIFAR-10**. The "step" is equivalent to the number of applying gradients. The global mini-batch size is increased by adding more workers $K$ with fixed $B_{\text{loc}} = 128$. Results are averaged over three runs, each with fine-tuned learning rate.

On contrary to standard large-batch training, *local SGD scales to larger batch size and provides additional parallelism upon the limitation of current large batch training.* Figure 9 shows the example of training ResNet-20 on CIFAR-10 with $H = 2$, which trains and generalizes better in terms of update steps while with reduced communication cost.

### B.4 PRACTICAL IMPROVEMENT POSSIBILITIES FOR STANDARD LOCAL SGD TRAINING

We investigate different aspects of the training to address the quality when scaling local SGD to the extreme case, e.g., hybrid momentum scheme, warming up the local SGD or fine-tuning the learning rate. In this section, we briefly present how these strategies are, and how they work in practice where we train ResNet-20 on CIFAR-10 on 16 GPUs.

### B.4.1 LOCAL SGD WITH MOMENTUM

Momentum mini-batch SGD is widely used in place of vanilla SGD. The distributed mini-batch SGD with vanilla momentum on $K$ training nodes follows

$$\boldsymbol{u}_{(t)} = m\boldsymbol{u}_{(t-1)} + \frac{1}{K}\sum_{k=1}^{K}\boldsymbol{\nabla}_{(t)}^{k}\,, \qquad \boldsymbol{w}_{(t+1)} = \boldsymbol{w}_{(t)} - \gamma\boldsymbol{u}_{(t)}$$

where $\boldsymbol{\nabla}_{(t)}^{k} = \frac{1}{|\mathcal{I}_{(t)}^{k}|}\sum_{i\in\mathcal{I}_{(t)}^{k}}\nabla f_i(\boldsymbol{w}_{(t)})$.

After $H$ updates of mini-batch SGD, we have the following updated $\boldsymbol{w}_{(t+H)}$:

$$\boldsymbol{w}_{(t+H)} = \boldsymbol{w}_{(t)} - \gamma\left(\sum_{\tau=1}^{H}m^{\tau}\boldsymbol{u}_{(t-1)} + \sum_{\tau=0}^{H-1}\frac{m^{\tau}}{K}\sum_{k=1}^{K}\boldsymbol{\nabla}_{(t)}^{k} + \ldots + \sum_{\tau=0}^{0}\frac{m^{\tau}}{K}\sum_{k=1}^{K}\boldsymbol{\nabla}_{(t+H-1)}^{k}\right)$$

Coming back to the setting of local SGD, we can apply momentum acceleration on each local model, or on a global level (Chen & Huo, 2016). In the remaining part of this section, we analyze the case of applying local momentum and global momentum. For ease of understanding, we assume the learning rate $\gamma$ is the same throughout the $H$ update steps.

**Local SGD with Local Momentum.** When applying local momentum on the local SGD, i.e., using independent identical momentum acceleration for each local model and only globally aggregating the

gradients at the time $(t) + H$, we have the following local update scheme

$$\boldsymbol{u}_{(t)}^k = m\boldsymbol{u}_{(t-1)}^k + \boldsymbol{\nabla}_{(t)}^k, \qquad \boldsymbol{w}_{(t)+1}^k = \boldsymbol{w}_{(t)}^k - \gamma\boldsymbol{u}_{(t)}^k,$$

where $\boldsymbol{\nabla}_{(t)}^k = \frac{1}{|\mathcal{I}_{(t)}^k|} \sum_{i \in \mathcal{I}_{(t)}^k} \nabla f_i(\boldsymbol{w}_{(t)})$. Consequently, after $H$ local steps,

$$\boldsymbol{w}_{(t)+H}^k = \boldsymbol{w}_{(t)}^k - \gamma\left(\sum_{\tau=1}^{H} m^\tau \boldsymbol{u}_{(t-1)}^k + \sum_{\tau=0}^{H-1} m^\tau \boldsymbol{\nabla}_{(t)}^k + \ldots + \sum_{\tau=0}^{0} m^\tau \boldsymbol{\nabla}_{(t)+H-1}^k\right).$$

Substituting the above equation into eq. (2), we have the update

$$\boldsymbol{w}_{(t+1)} = \boldsymbol{w}_{(t)} - \frac{1}{K}\sum_{k=1}^{K} \gamma\left(\sum_{\tau=1}^{H} m^\tau \boldsymbol{u}_{(t-1)}^k \sum_{\tau=0}^{H-1} m^\tau \boldsymbol{\nabla}_{(t)}^k + \ldots + \sum_{\tau=0}^{0} m^\tau \boldsymbol{\nabla}_{(t)+H-1}^k\right)$$

$$= w_{(t)} - \gamma\left(\sum_{\tau=1}^{H} \frac{m^\tau}{K}\sum_{k=1}^{K} \boldsymbol{u}_{(t-1)}^k + \sum_{\tau=0}^{H-1} m^\tau(\frac{1}{K}\sum_{k=1}^{K}\boldsymbol{\nabla}_{(t)}^k) + \ldots + \sum_{\tau=0}^{0}\frac{m^\tau}{K}\sum_{k=1}^{K}\boldsymbol{\nabla}_{(t)+H-1}^k\right)$$

Comparing the mini-batch SGD with local momentum local SGD after $H$ update steps ($H$ global update steps v.s. $H$ local update steps and 1 global update step), we witness that the main difference of these two update schemes is the difference between $\sum_{\tau=1}^{H} m^\tau \boldsymbol{u}_{(t-1)}$ and $\sum_{\tau=1}^{H}\frac{m^\tau}{K}\sum_{k=1}^{K}\boldsymbol{u}_{(t-1)}^k$, where mini-batch SGD holds a global $\boldsymbol{u}_{(t-1)}$ while each local model of the local SGD has their own $\boldsymbol{u}_{(t-1)}^k$. We will soon see the difference between the global momentum of mini-batch SGD and the local momentum of local SGD.

**Local SGD with Global Momentum** For global momentum local SGD, i.e., a more general variant of block momentum (Chen & Huo, 2016), we would like to apply the momentum factor only to the accumulated/synchronized gradients:

$$\boldsymbol{u}_{(t)} = m\boldsymbol{u}_{(t-1)} + \frac{1}{\gamma}\sum_{k=1}^{K}\frac{1}{K}(\boldsymbol{w}_{(t)}^k - \boldsymbol{w}_{(t)+H}^k) = m\boldsymbol{u}_{(t-1)} + \frac{1}{\gamma}\sum_{k=1}^{K}\frac{1}{K}\sum_{l=0}^{H-1}\gamma\boldsymbol{\nabla}_{(t)+l}^k,$$

$$\boldsymbol{w}_{(t+1)} = \boldsymbol{w}_{(t)} - \gamma\boldsymbol{u}_{(t)} = \boldsymbol{w}_{(t)} - \gamma\big(m\boldsymbol{u}_{(t-1)} + \sum_{l=0}^{H-1}\sum_{k=1}^{K}\frac{1}{K}\boldsymbol{\nabla}_{(t)+l}^k\big)$$

where $\boldsymbol{w}_{(t)+H}^k = \boldsymbol{w}_{(t)}^k - \eta\sum_{l=0}^{H-1}\boldsymbol{\nabla}_{(t)+l}^k = \boldsymbol{w}_{(t)} - \eta\sum_{l=0}^{H-1}\boldsymbol{\nabla}_{(t)+l}^k$. Note that for local SGD, we consider summing up the gradients from each local update, i.e., the model difference before and after one global synchronization, and then apply the global momentum to the gradients over workers over previous local update steps.

Obviously, there exists a significant difference between mini-batch momentum SGD and global momentum local SGD, at least the term $\sum_{\tau=0}^{H} m^\tau$ is cancelled.

**Local SGD with Hybrid Momentum.** The following equation tries to combine local momentum with global momentum, showing a naive implementation.

First of all, based on the local momentum scheme, after $H$ local update steps,

$$\boldsymbol{w}_{(t)+H}^k = \boldsymbol{w}_{(t)}^k - \gamma\big(\sum_{\tau=1}^{H} m^\tau \boldsymbol{u}_{(t-1)}^k + \sum_{\tau=0}^{H-1} m^\tau \boldsymbol{\nabla}_{(t)}^k + \ldots + \sum_{\tau=0}^{0} m^\tau \boldsymbol{\nabla}_{(t)+H-1}^k\big)$$

Together with the result from local momentum with the global momentum, we have

$$\boldsymbol{u}_{(t)} = m\boldsymbol{u}_{(t-1)} + \frac{1}{\gamma}\sum_{k=1}^{K}\frac{1}{K}(\boldsymbol{w}_{(t)}^k - \boldsymbol{w}_{(t)+H}^k)$$

$$\boldsymbol{w}_{(t+1)} = \boldsymbol{w}_{(t)} - \gamma\boldsymbol{u}_{(t)} = \boldsymbol{w}_{(t)} - \gamma\left[m\boldsymbol{u}_{(t-1)} + \frac{1}{\gamma}\sum_{k=1}^{K}\frac{1}{K}(\boldsymbol{w}_{(t)}^k - \boldsymbol{w}_{(t)+H}^k)\right]$$

$$= \boldsymbol{w}_{(t)} - \gamma\left[m\boldsymbol{u}_{(t-1)} + \sum_{\tau=1}^{H}\frac{m^\tau}{K}\sum_{k=1}^{K}\boldsymbol{u}_{(t-1)}^k + \sum_{\tau=0}^{H-1}\frac{m^\tau}{K}\sum_{k=1}^{K}\boldsymbol{\nabla}_{(t)}^k + \ldots + \sum_{\tau=0}^{0}\frac{m^\tau}{K}\sum_{k=1}^{K}\boldsymbol{\nabla}_{(t)+H-1}^k\right]$$

where $\boldsymbol{u}_{(t-1)}$ is the global momentum memory and $\boldsymbol{u}_{(t-1)}$ is the local momentum memory for each node $k$.

Table 8: Evaluate local momentum and global momentum for **ResNet-20** on **CIFAR-10** data via local SGD training ($H = 1$ case) on $5 \times 2$-GPU Kubernetes cluster. The local mini-batch size is 128 and base batch size is 64 (used for learning rate linear scale). Each local model will access to a disjoint data partition, using the standard learning rate scheme as He et al. (2016a).

| local momentum | global momentum | test top-1 |
|---|---|---|
| 0.0 | 0.0 | 90.57 |
| 0.9 | 0.0 | 92.41 |
| 0.9 | 0.1 | 92.22 |
| 0.9 | 0.2 | 92.09 |
| 0.9 | 0.3 | 92.54 |
| 0.9 | 0.4 | 92.45 |
| 0.9 | 0.5 | 92.19 |
| 0.9 | 0.6 | 91.32 |
| 0.9 | 0.7 | 18.76 |
| 0.9 | 0.8 | 14.35 |
| 0.9 | 0.9 | 12.21 |
| 0.9 | 0.95 | 10.11 |

**Local SGD with Momentum in Practice.** In practice, it is possible to combine the local momentum with global momentum to further improve the model performance. For example, a toy example in Table 8 investigates the impact of different momentum schemes on CIFAR-10 trained with ResNet-20 on a $5 \times 2$-GPU cluster, where some factors of global momentum could further slightly improve the final test accuracy.

However, the theoretical understanding of how local momentum and global momentum contribute to the optimization still remains unclear, which further increase the difficulty of tuning local SGD over $H$, $K$. An efficient way of using local and global momentum remains a future work and in this work, we only consider the local momentum.

### B.4.2    WARM-UP OF THE NUMBER OF LOCAL SGD STEPS

We use the term "local step warm-up strategy", to refer to a specific variant of post-local SGD. More precisely, instead of the two-phase regime which we presented here the used number of local steps $H$ will be gradually increased from 1 to the expected number of local steps $H$. The warm-up strategies investigated here are "linear", "exponential" and "constant".

Please note that the implemented post-local SGD over the whole text only refers to the training scheme that uses frequent communication (i.e., $H = 1$) before the first learning rate decay and then reduces the communication frequency (i.e., $H > 1$) after the decay.

This section then investigates the trade-off between stochastic noise and the training stability. Also note that the Figure 2(a) in the main text has already presented one aspect of the trade-off. So the exploration below mainly focuses on the other aspects and tries to understand how will the scale of the added stochastic noise impact the training stability. Infact, even the model has been stabilized to a region with good quality.

Figure 10 and Figure 11 investigate the potential improvement through using the local step warm-up strategy for the case of training ResNet-20 on CIFAR-10. Figure 10 evaluates the warm-up strategies of "linear" and "constant" for different $H$, while the evaluation of "exponential" warm-up strategy is omitted due to its showing similar performance as "linear" warm-up.

However, none of the investigated strategies show convincing performance. Figure 11 further studies how the period of warm-up impacts the training performance. We can witness that even if we increase the warm-up phase to 50 epochs where the training curve of mini-batch SGD becomes stabilized, the

large noise introduced by the local SGD will soon degrade the status of training and lead to potential quality loss, as in Figure 11(a).

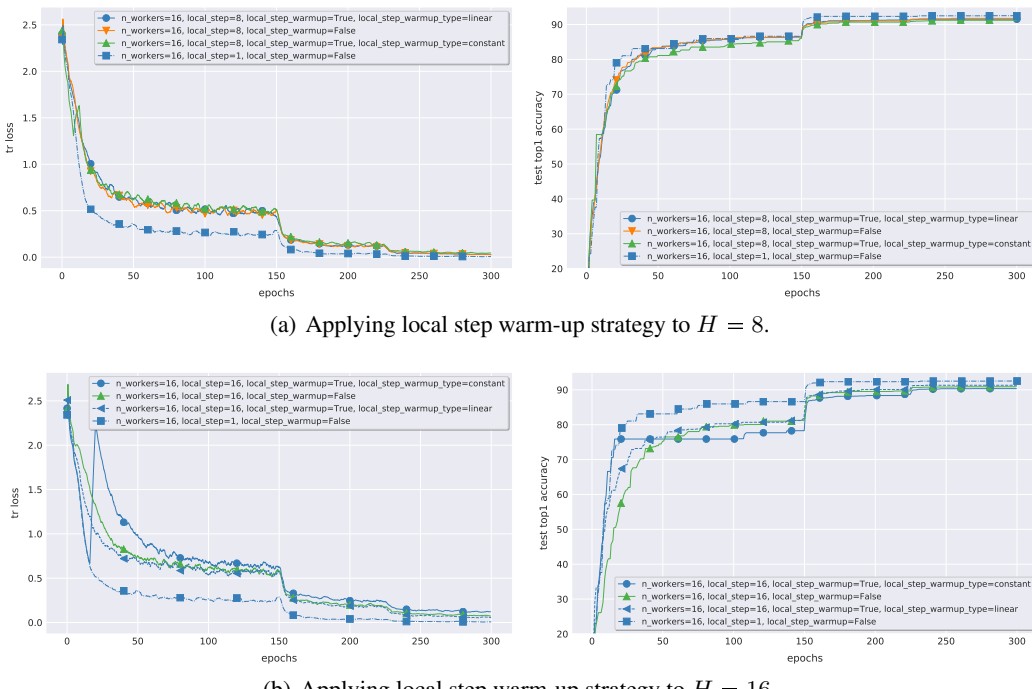

(a) Applying local step warm-up strategy to $H = 8$.

(b) Applying local step warm-up strategy to $H = 16$.

Figure 10: Investigate how local step warm-up strategy impacts the performance of training **CIFAR-10** with **ResNet-20** via **local SGD** ($8 \times 2$-GPU). The local batch size $B_{\text{loc}}$ is fixed to 128. The warmup strategies are "linear" and "constant", and the warm-up period used here is equivalent to the number of local steps $H$.

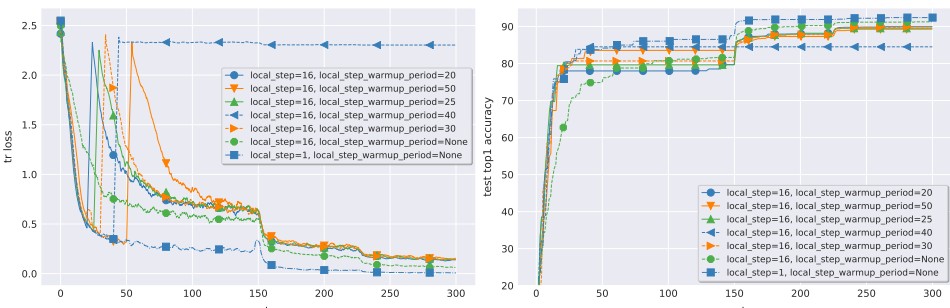

(a) Evaluate the impact of "constant" local step warm-up for different period of warm-up phase.

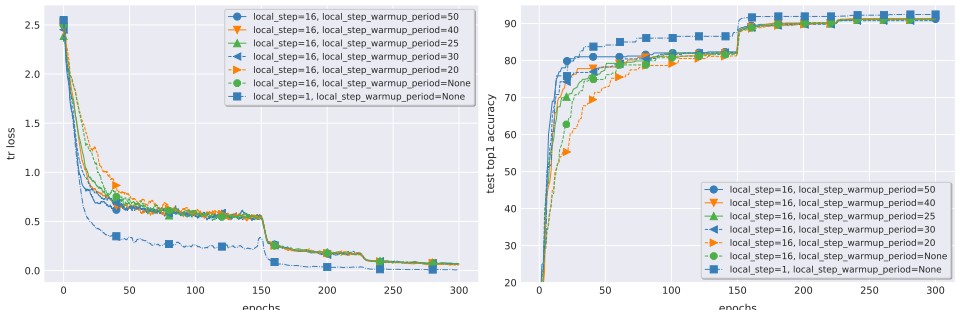

(b) Evaluate the impact of "linear" local step warm-up for different period of warm-up phase.

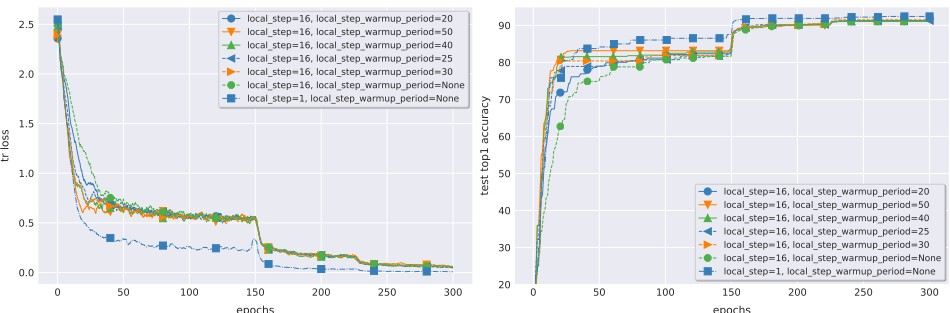

(c) Evaluate the impact of "exponential" local step warm-up for different period of warm-up phase.

Figure 11: Investigate how different warm-up period of the local step warm-up impacts the performance of training **CIFAR-10** with **ResNet-20** via **local SGD** ($8 \times 2$-GPU). The local batch size $B_{\text{loc}}$ is fixed to 128, and the strategies to warm-up the number of local steps $H$ are "linear", "exponential" and "constant".

## C  POST-LOCAL SGD TRAINING

### C.1  THE ALGORITHM OF POST-LOCAL SGD

---

**Algorithm 2** *Post-local SGD*

---

**input:** the initial model $\boldsymbol{w}_{(0)}$;
**input:** training data with labels $\mathcal{I}$;
**input:** mini-batch of size $B_{\text{loc}}$ per local model;
**input:** step size $\eta$, and momentum $m$ (optional);
**input:** number of synchronization steps $T$, and the first learning rate decay is performed at $T'$;
**input:** number of eventual local steps $H'$;
**input:** number of nodes $K$.
 1: synchronize to have the same initial models $\boldsymbol{w}_{(0)}^k := \boldsymbol{w}_{(0)}$.
 2: **for all** $k := 1, \ldots, K$ **do in parallel**
 3:    **for** $t := 1, \ldots, T$ **do**
 4:       **if** $t < T'$ **then**
 5:          $H_{(t)} = 1$
 6:       **else**
 7:          $H_{(t)} = H'$
 8:       **end if**
 9:       **for** $h := 1, \ldots, H_{(t)}$ **do**
10:          sample a mini-batch from $\mathcal{I}_{(t)+h-1}^k$.
11:          compute the gradient

$$\boldsymbol{g}_{(t)+h-1}^k := \tfrac{1}{B_{\text{loc}}} \sum_{i \in \mathcal{I}_{(t)+h-1}^k} \nabla f_i\left(\boldsymbol{w}_{(t)+h-1}^k\right).$$

12:          update the local model to

$$\boldsymbol{w}_{(t)+h}^k := \boldsymbol{w}_{(t)+h-1} - \gamma_{(t)} \boldsymbol{g}_{(t)+h-1}^k.$$

13:       **end for**
14:       all-reduce aggregation of the gradients

$$\Delta_{(t)}^k := \boldsymbol{w}_{(t)}^k - \boldsymbol{w}_{(t)+H}^k.$$

15:       get new global (synchronized) model $\boldsymbol{w}_{(t+1)}^k$ for all $K$ nodes:

$$\boldsymbol{w}_{(t+1)}^k := \boldsymbol{w}_{(t)}^k - \gamma_{(t)} \tfrac{1}{K} \sum_{i=1}^K \Delta_{(t)}^k$$

16:    **end for**
17: **end for**

---

## C.2   The Effectiveness of Turning on Post-local SGD after the First Learning Rate Decay

In Figure 12, we study the sufficiency as well as the necessity of "injecting" more stochastic noise (i.e., using post-local SGD) into the optimization procedure after performing the first learning rate decay. Otherwise, the delayed noise injection (i.e., starting the post-local SGD only from the second learning rate decay) not only introduces more communication cost but also meets the increased risk of converging to sharper minima.

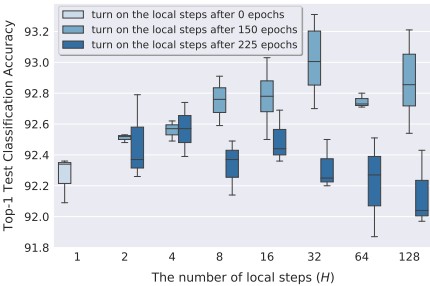

Figure 12:  The effectiveness and necessary of turning on the post-local SGD after the first learning rate decay. The example here trains **ResNet-20** on **CIFAR-10** on $K = 16$ GPUs with $B_{\text{loc}} K = 2048$.

## C.3   The Speedup of Post-local SGD Training on CIFAR

Table 3 and Table 10 evaluate the speedup of mini-batch SGD and post-local SGD, over different CNN models, datasets, and training phases.

Table 9:  The **Speedup** of mini-batch SGD and post-local SGD, over different CNN models and datasets. The speedup is evaluated by $\frac{T_a}{T_a^K}$, where $T_a$ is the training time of the algorithm $a$ on 1 GPU and $T_a^K$ corresponds to the training on $K$ GPUs. We use 16 GPUs in total (with $B_{\text{loc}} = 128$) on an $8 \times 2$-GPU cluster with 10 Gbps network bandwidth. The experimental setup is the same as Table 3.

|                  | CIFAR-10 | | | CIFAR-100 | | |
|------------------|------|-------|-------|------|-------|-------|
|                  | H=1  | H=16  | H=32  | H=1  | H=16  | H=32  |
| ResNet-20        | 9.45 | 12.24 | 13.00 | 8.75 | 11.05 | 11.67 |
| DenseNet-40-12   | 8.31 | 10.80 | 11.37 | 8.04 | 10.59 | 10.85 |
| WideResNet-28-12 | 5.33 | 7.94  | 8.19  | 5.29 | 7.83  | 8.14  |

Table 10:  The **Speedup** of mini-batch SGD and post-local SGD (only consider the phase of performing post-local SGD) over different CNN models and datasets. The speedup is evaluated by $\frac{T_a}{T_a^K}$, where $T_a$ is the training time of the algorithm $a$ (corresponding to the second phase) on 1 GPU and $T_a^K$ corresponds to the training on $K$ GPUs. We use 16 GPUs in total (with $B_{\text{loc}} = 128$) on an $8 \times 2$-GPU cluster with 10 Gbps network bandwidth. The experimental setup is the same as Table 3.

|                  | CIFAR-10 | | | CIFAR-100 | | |
|------------------|------|-------|-------|------|-------|-------|
|                  | H=1  | H=16  | H=32  | H=1  | H=16  | H=32  |
| ResNet-20        | 9.45 | 17.33 | 20.80 | 8.75 | 15.00 | 17.50 |
| DenseNet-40-12   | 8.31 | 15.43 | 18.00 | 8.04 | 15.50 | 16.69 |
| WideResNet-28-12 | 5.33 | 15.52 | 17.66 | 5.29 | 15.09 | 17.69 |

## C.4 Understanding the Generalization of Post-local SGD

**The Sharpness Visualization**   Figure 13 visualizes the sharpness of the minima for training ResNet-20 on CIFAR-10. 10 different random direction vectors are used for the filter normalization (Li et al., 2018), to ensure the correctness and consistence of the sharp visualization.

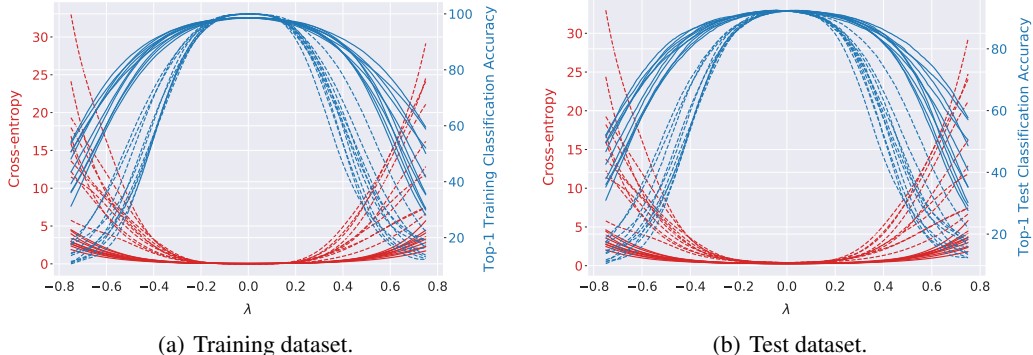

(a) Training dataset.                                      (b) Test dataset.

Figure 13: Sharpness visualization of the minima for **ResNet-20** trained on **CIFAR-10**. The training is on top of $K = 16$ GPUs and the local batch size is fixed to $B_{\text{loc}} = 128$. The dashed lines are standard mini-batch SGD and the solid lines are post-local SGD with $H = 16$. The sharpness visualization of minima is performed via filter normalization (Li et al., 2018). The model is perturbed as $w + \lambda d$ by a shared random direction $d$, and is evaluated by the whole dataset (training or test respectively). The top-1 test accuracy of mini-batch SGD is 92.25, while that of post-local SGD is 92.61. The sharpness of these two minima is consistent over 10 different random directions.

**The spectrum of the Hessian for mini-batch SGD and post-local SGD**   Figure 14 evaluates the spectrum of the Hessian for the model trained from mini-batch SGD and post-local SGD with different $H$, which again demonstrates the fact that large-batch SGD tends to stop at points with high Hessian spectrum while post-local SGD could easily generalize to a low curvature solution and with better generalization.

**1-d linear interpolation between models**   The 1-d linear interpolation was first used by Goodfellow et al. (2015) and then widely used to study the "sharpness" and "flatness" of different minima in several works (Keskar et al., 2017; Dinh et al., 2017; Li et al., 2018). In Figure 15, the model trained by post-local SGD ($w_{\text{post-local SGD}}$) can generalize to flatter minima than that of mini-batch SGD ($w_{\text{mini-batch SGD}}$), either $w_{\text{post-local SGD}}$ is trained from scratch, or resumed from the checkpoint of $w_{\text{mini-batch SGD}}$ (i.e., the checkpoint is one-epoch ahead of the first learning rate decay so as to share the common weight structure with $w_{\text{mini-batch SGD}}$).

## C.5 Post-local SGD Training on Diverse Tasks

### C.5.1 Post-local SGD Training on CIFAR-100 for Global Mini-batch Size $KB_{\text{LOC}} = 4096$

Table 11 presents the severe quality loss (at around $2\%$) of the fine-tuned large-batch SGD for training three CNNs on CIFAR-100. Our post-local SGD with default hyper-parameters (i.e., the hyper-parameters from small mini-batch size and via large-batch training schemes) can perfectly close the generalization gap or even better the fine-tuned small mini-batch baselines.

We further justify the argument of works (Hoffer et al., 2017; Shallue et al., 2018) in Table 12, where we increase the number of training epochs and train it longer (from 300 to 400 and 500) for ResNet-20 on CIFAR-100. The results below illustrate that increasing the number of training epochs alleviates the optimization difficulty of large-batch training

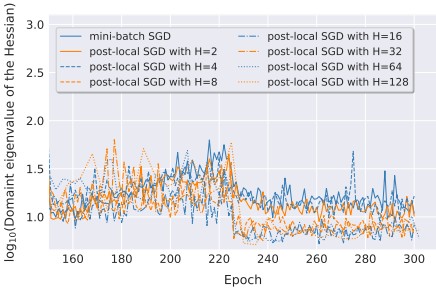

(a) The dominant eigenvalue of the Hessian, which is evaluated on the training dataset per epoch. Only the phase related to the strategy of post-local SGD is visualized. Mini-batch SGD or post-local SGD with very much $H$ (e.g., $H = 2, 4$) have noticeably larger dominant eigenvalue.

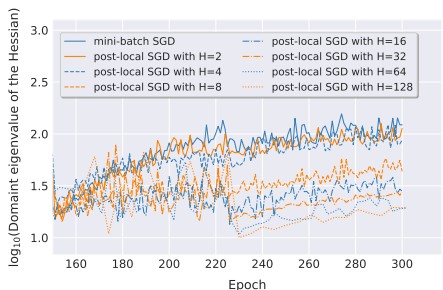

(b) The dominant eigenvalue of the Hessian, which is evaluated on the test dataset per epoch. Only the phase related to the strategy of post-local SGD is visualized. Mini-batch SGD or post-local SGD with very much $H$ (e.g., $H = 2, 4$) have noticeably larger dominant eigenvalue.

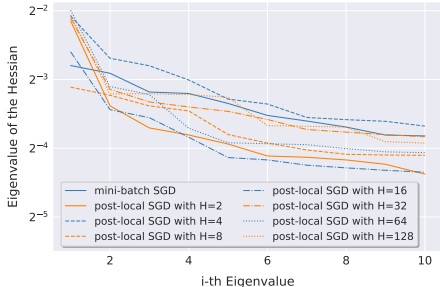

(c) Top 10 eigenvalues of the Hessian, which is evaluated on the training dataset for the best model of each training scheme. The top eigenvalues of the Hessian of the mini-batch SGD clearly present significant different pattern than the post-local SGD counterpart.

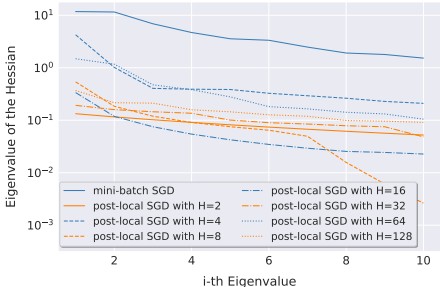

(d) Top 10 eigenvalues of the Hessian, which is evaluated on the test dataset for the best model of each training scheme. The top eigenvalues of the Hessian of the mini-batch SGD clearly present significant different pattern than the post-local SGD counterpart.

Figure 14: The spectrum of the Hessian for **ResNet-20** trained on **CIFAR-10**. The training is on top of $K = 16$ GPUs with $KB_{\mathrm{loc}} = 2048$. The spectrum is computed using power iteration (Martens & Sutskever, 2012; Yao et al., 2018) with the relative error of 1e-4. The top-1 test accuracy of mini-batch SGD is 92.57, while that of post-local SGD ranges from 92.33 to 93.07. Current large-batch SGD tends to stop at points with considerably "larger" Hessian spectrum, while large-batch trained with post-local SGD generalizes to solution with low curvature and with better generalization.

Table 11: Top-1 **test accuracy** of training different CNN models via **post-local SGD** on $K = 32$ GPUs with a large batch size ($B_{\mathrm{loc}}K = 4096$). The reported results are the average of three runs. We include the small and large batch baseline, where the models are trained by mini-batch SGD with mini-batch size 256 and 4096 respectively. The $\star$ indicates the fine-tuned learning rate.

| | CIFAR-100 | | | |
|---|---|---|---|---|
| | small batch baseline $\star$ | large batch baseline $\star$ | post-local SGD (H=8) | post-local SGD (H=16) |
| **ResNet-20** | 68.84 $\pm 0.06$ | 67.34 $\pm 0.34$ | 68.38 $\pm 0.48$ | 68.30 $\pm 0.30$ |
| **DenseNet-40-12** | 74.85 $\pm 0.14$ | 73.00 $\pm 0.04$ | 74.50 $\pm 0.34$ | 74.96 $\pm 0.30$ |
| **WideResNet-28-10** | 79.78 $\pm 0.16$ | 77.82 $\pm 0.65$ | 79.53 $\pm 0.45$ | 79.80 $\pm 0.39$ |

### C.5.2 POST-LOCAL SGD TRAINING ON LANGUAGE MODELING

We evaluate the effectiveness of post-local SGD for training the language modeling task on WikiText-2 through LSTM. We borrowed and adapted the general experimental setup of Merity et al. (2018), where we use a three-layer LSTM with hidden dimension of size 650. The loss will be averaged over all examples and timesteps. The BPTT length is set to 30. We fine-tune the value of gradient clipping (0.4) and the dropout (0.4) is only applied on the output of LSTM. The local mini-batch size $B_{\mathrm{loc}}$ is

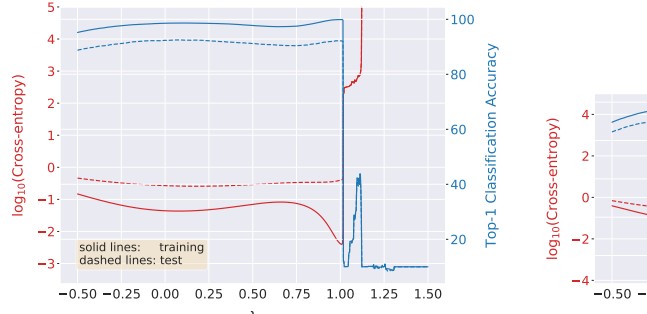
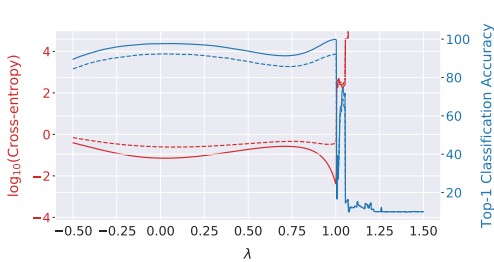

(a) $\boldsymbol{w}_{\text{post-local SGD}}$ is trained with $H = 16$, where the training resumes from the checkpoint of $\boldsymbol{w}_{\text{mini-batch SGD}}$ which is one-epoch ahead of the first learning rate decay.

(b) $\boldsymbol{w}_{\text{post-local SGD}}$ is trained with $H = 32$, where the training resumes from the checkpoint of $\boldsymbol{w}_{\text{mini-batch SGD}}$ which is one-epoch ahead of the first learning rate decay.

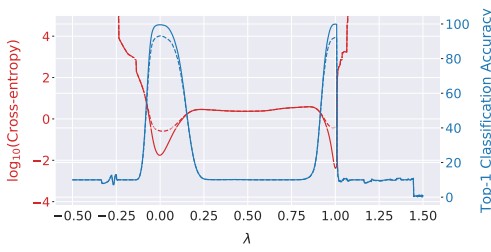
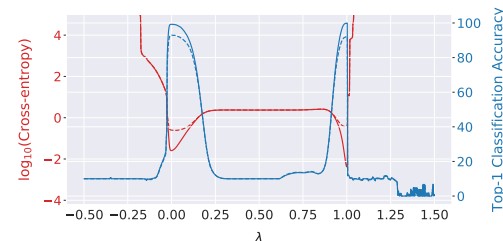

(c) $\boldsymbol{w}_{\text{post-local SGD}}$ is trained from scratch with $H = 16$.

(d) $\boldsymbol{w}_{\text{post-local SGD}}$ is trained from scratch with $H = 32$.

Figure 15: 1-d linear interpolation between models $\boldsymbol{w}_{\text{post-local SGD}}$ and $\boldsymbol{w}_{\text{mini-batch SGD}}$, i.e., $\hat{\boldsymbol{w}} = \lambda \boldsymbol{w}_{\text{mini-batch SGD}} + (1 - \lambda)\boldsymbol{w}_{\text{post-local SGD}}$, for different minima of **ResNet-20** trained on **CIFAR-10**. The training is on top of $K = 16$ GPUs and the local batch size is fixed to $B_{\text{loc}} = 128$. The solid lines correspond to evaluate $\hat{\boldsymbol{w}}$ on the whole training dataset while the dashed lines are on the test dataset. The post-local SGD in Figure 15(a) and Figure 15(d) is trained from the checkpoint of $\boldsymbol{w}_{\text{mini-batch SGD}}$ before performing the first learning rate decay, while that of Figure 15(c) and Figure 15(d) is trained from scratch. The top-1 test accuracy of mini-batch SGD is 92.25, while that of post-local SGD in Figure 15(a), Figure 15(b), Figure 15(c) and Figure 15(d), are 92.61, 92.35, 93.13 and 93.04 respectively.

Table 12: Top-1 **test accuracy** of large-batch SGD and post-local SGD, for training ResNet-20 on CIFAR-100 with $B_{\text{loc}}K = 4096$. The reported results are the average of three runs. We include the small and large batch baseline, where the models are trained by mini-batch SGD with 256 and 4096 respectively. The $\star$ indicates the fine-tuned learning rate. The learning rate will be decayed by 10 when the distributed algorithm has accesses 50% and 75% of the total number of training samples.

| # of epochs | small batch baseline $\star$ | large batch baseline $\star$ | post-local SGD (H=8) | post-local SGD (H=16) |
|---|---|---|---|---|
| **300** | 68.84 $\pm 0.06$ | 67.34 $\pm 0.34$ | 68.38 $\pm 0.48$ | 68.30 $\pm 0.30$ |
| **400** | 69.07 $\pm 0.27$ | 67.55 $\pm 0.21$ | 69.06 $\pm 0.15$ | 69.05 $\pm 0.26$ |
| **500** | 69.03 $\pm 0.10$ | 67.42 $\pm 0.63$ | 69.02 $\pm 0.38$ | 68.87 $\pm 0.27$ |

64 and we train the model for 120 epochs. The learning rate is again decayed at the phase when the training algorithm has accessed 50% and 75% of the total training samples.

Table 13 below demonstrates the effectiveness of post-local SGD for large-batch training on language modeling task (without extra fine-tuning except our baselines). Note that most of the existing work focuses on improving the large-batch training issue for computer vision tasks; and it is non-trivial to scale the training of LSTM for language modeling task due to the presence of different hyper-parameters. Here we provide a proof of concept result in Table 13 to show that our post-local SGD is able to improve upon standard large-batch baseline. The benefits can be further pronounced if we scale to larger batches (as the case of image classification e.g. in Table 11 and Table 12).

Table 13: The perplexity (lower is better) of language modeling task on WikiText-2. We use $K = 16$ and $KB = KB_{\text{loc}} = 1024$. The reported results are evaluated on the validation dataset (average of three runs). We fine-tune the learning rate for mini-batch SGD baselines.

| small batch baseline ⋆ | large batch baseline ⋆ | large-batch (H=8) | large-batch (H=16) |
|---|---|---|---|
| 86.50 ±0.35 | 86.90 ±0.49 | 86.61 ±0.30 | 86.85 ±0.13 |

### C.5.3 POST-LOCAL SGD TRAINING ON IMAGENET

We evaluate the performance of post-local SGD on the challenging ImageNet training. Again we limit ResNet-50 training to 90 passes over the data in total, and use the standard training configurations as mentioned in Appendix A.4.2. The post-local SGD begins when performing the first learning rate decay.

We can witness that post-local SGD outperforms mini-batch SGD baseline for both of mini-batch size 4096 (76.18 and 75.87 respectively) and 8192 (75.65 and 75.64 respectively).

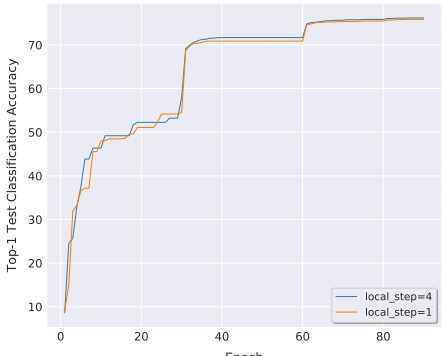

(a) The performance of post-local SGD training for **ImageNet-1k** with $KB_{\text{loc}} = 4096$, in terms of epoch-to-accuracy.

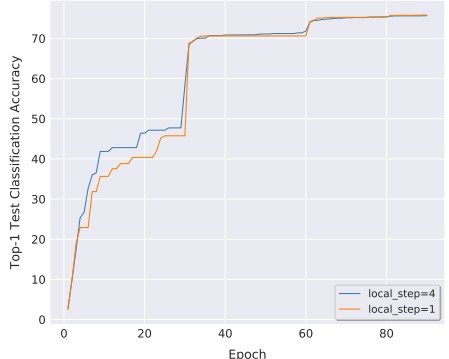

(b) The performance of post-local SGD training for **ImageNet-1k** with $KB_{\text{loc}} = 8192$, in terms of epoch-to-accuracy.

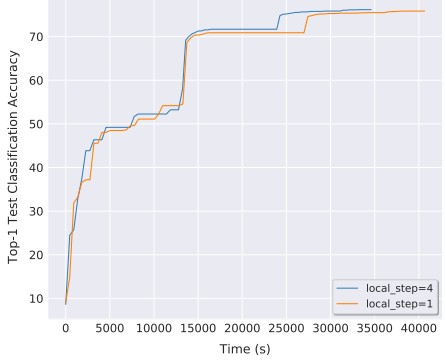

(c) The performance of post-local SGD training for **ImageNet-1k** with $KB_{\text{loc}} = 4096$, in terms of time-to-accuracy.

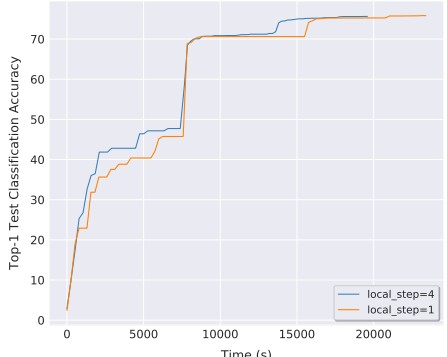

(d) The performance of post-local SGD training for **ImageNet-1k** with $KB_{\text{loc}} = 8192$, in terms of time-to-accuracy.

Figure 16: The performance of **post-local SGD** training for **ImageNet-1k**. We evaluate the model performance on test dataset after each complete accessing of the whole training samples. Note that due to the resource limitation of the main experimental platform in the paper, these experiments are on top of a $8 \times 4$-GPU (V100) Kubernetes cluster with 10 Gbps network bandwidth.

### C.5.4 POST-LOCAL SGD VS. OTHER NOISE INJECTION METHODS

The role of "noise" has been actively studied in SGD for non-convex deep learning, from optimization and generalization aspects. Neelakantan et al. (2015) propose to inject isotropic white noise for better

optimization. Zhu et al. (2019); Xing et al. (2018) study the "structured" anisotropic noise and find the importance of anisotropic noise in SGD for escaping from minima (over isotropic noise) in terms of generalization. Wen et al. (2019) on top of these work and try to inject noise (sampled from the expensive empirical Fisher matrix) to large-batch SGD.

However, to our best knowledge, none of the prior work can provide a computation efficient way to inject noise to achieve as good generalization performance as small-batch SGD. Either it is practically unknown if injecting the isotropic noise (Neelakantan et al., 2015) can alleviate the issue of large-batch training, or it has been empirically justified (Wen et al., 2019) that injecting anisotropic noise from expensive (diagonal) empirical Fisher matrix fails to recover the same performance as the small mini-batch baselines. In this section, we only evaluate the impact of injecting isotropic noise (Neelakantan et al., 2015) for large-batch training, and omit the comparison to Wen et al. (2019)[8].

The idea of Neelakantan et al. (2015) considers to add time-dependent Gaussian noise to the gradient at every training step $t$: $\nabla f_i(\boldsymbol{w}_{(t)}) \leftarrow \nabla f_i(\boldsymbol{w}_{(t)}) + \mathcal{N}(0, \boldsymbol{\sigma}_t^2)$, where $\boldsymbol{\sigma}_t^2$ follows $\boldsymbol{\sigma}_t^2 := \frac{\eta}{(1+t)^{\gamma'}}$. We follow the general hyper-parameter search scheme (as mentioned in Section A.4) and fine-tune the $\eta$ and $\gamma'$ in the range of $\{1e^{-6}, 5e^{-6}, 1e^{-5}, 5e^{-5}\}$ and $\{0.5, 1, 1.5, 2, 2.5, 3.0\}$ respectively.

Table 14 below compares the post-local SGD to Neelakantan et al. (2015), where the noise injection scheme in Neelakantan et al. (2015) cannot address the large-batch training issue and could even deteriorate the generalization performance. Note that we also tried to search the hyper-parameters at around the values reported in Neelakantan et al. (2015) for our state-of-the-art CNNs, but it will directly result in the severe quality loss or even divergence.

Table 14: Top-1 **test accuracy** of adding isotropic noise for large-batch training. We revisit the case of ResNet-20 in Table 3 (2K for CIFAR-10) and Table 11 (4K for CIFAR-100), where adding isotropic noise as in Neelakantan et al. (2015) cannot address the large-batch training difficulty. The $\star$ indicates a fine-tuned learning rate.

|  | **CIFAR-10**, $KB = 2K$ | **CIFAR-100**, $KB = 4K$ |
|---|---|---|
| Neelakantan et al. (2015)$\star$ | 92.46 $\pm 0.27$ | 67.15 $\pm 0.26$ |
| **mini-batch SGD** $\star$ | 92.48 $\pm 0.17$ | 67.38 $\pm 0.34$ |
| **Post-local SGD** | **93.02** $\pm 0.24$ | **68.30** $\pm 0.48$ |

**Compared to another local SGD variant.** Recently, Wang & Joshi (2019) proposed to decrease the number of local update steps $H$ during training. However, their scheme is inherited from the convergence analysis for optimization (not the generalization for deep learning), and in principle opposite to our interpolation and proposed strategy (i.e. increasing the local update steps during the training).

Their evaluation (ResNet-50 on CIFAR-10 for $K = 4$ with $B_{loc} = 128$) also does not cover the difficult large batch training scenario (Scenario 2), e.g., $H = 16, K = 16, B_{loc} = 128$. For the same CIFAR-10 task and $K = 4$ as in Wang & Joshi (2019), our smaller ResNet-20 with local SGD can simply reach a better accuracy with less communication[9] (Figure 2).

### C.5.5 Post-local SGD with Other Compression Schemes

In this subsection, we demonstrate that (post-)local SGD can be integrated with other compression techniques for better training efficiency (further reduced communication cost) and improved generalization performance (w.r.t. the gradient compression methods).

---

[8]On the one hand, it is non-trivial to re-implement their method due to the the unavailable code. On the other hand, it is known that our post-local SGD outperforms their expensive noise injection scheme (Wen et al., 2019), based on the comparison of their Table 1 and our Table 11-12.

[9]We directly compare to the reported values of Wang & Joshi (2019), as some missing descriptions and hyperparameters prevented us from reproducing the results ourselves. $H$ in their paper forms a decreasing sequence starting with 10 while our local SGD uses constant $H = 16$ during training.

We use sign-based compression scheme (i.e. signSGD (Bernstein et al., 2018) and EF-signSGD (Karimireddy et al., 2019)) for the demonstration. The pseudo code can be found in Algorithm 3 and Algorithm 4.

**Post-local SGD with signSGD.** We slightly adapt the original signSGD (Bernstein et al., 2018) to fit in the local SGD framework. The local model will be firstly updated by the sign of the local update directions (e.g. gradients, or gradients with weight decay and momentum acceleration), and then be synchronized to reach the global consensus model for the next local updates.

Note that we can almost recover the signSGD in Bernstein et al. (2018) from Algorithm 3 when $H' = 1$, except that we will average over the sign instead of using the majority vote in Bernstein et al. (2018). Table 15 illustrates the trivial generalization performance difference of these two schemes, where the post-local SGD in Algorithm 3 is able to significantly improve the generalization performance (as in Table 4) with further improved communication efficiency.

---

**Algorithm 3** *(Post-)Local SGD with the compression scheme in signSGD*

---

**input:** the initial model $\boldsymbol{w}_{(0)} \in \mathbb{R}^d$; training data with labels $\mathcal{I}$; mini-batch of size $B_{\text{loc}}$ per local model; step size $\eta$, and momentum $\boldsymbol{m}$ (optional); number of synchronization steps $T$, and the first learning rate decay is performed at $T'$; number of eventual local steps $H'$; number of nodes $K$.

1: synchronize to have the same initial models $\boldsymbol{w}_{(0)}^k := \boldsymbol{w}_{(0)}$.
2: **for all** $k := 1, \ldots, K$ **do in parallel**
3:     **for** $t := 1, \ldots, T$ **do**
4:         **if** $t < T'$ **then**
5:             $H_{(t)} = 1$
6:         **else**
7:             $H_{(t)} = H'$
8:         **end if**
9:         **for** $h := 1, \ldots, H_{(t)}$ **do**
10:             sample a mini-batch from $\mathcal{I}_{(t)+h-1}^k$
11:             compute the gradients $\boldsymbol{g}_{(t)+h-1}^k := \frac{1}{B_{\text{loc}}} \sum_{i \in \mathcal{I}_{(t)+h-1}^k} \nabla f_i\left(\boldsymbol{w}_{(t)+h-1}^k\right)$.     $\triangleright$ can involve weight decay and momentum.
12:             update the local model $\boldsymbol{w}_{(t)+h}^k := \boldsymbol{w}_{(t)+h-1} - \gamma_{(t)} \text{sign}(\boldsymbol{g}_{(t)+h-1}^k)$.
13:         **end for**
14:         get model difference $\Delta_{(t)}^k := \boldsymbol{w}_{(t)}^k - \boldsymbol{w}_{(t)+H}^k$.
15:         compress the model difference: $\boldsymbol{s}_{(t)}^k = \text{sign}(\Delta_{(t)}^k)$ and $\boldsymbol{p}_{(t)}^k = \frac{\left\|\Delta_{(t)}^k\right\|_1}{d}$.
16:         get new global (synchronized) model $\boldsymbol{w}_{(t+1)}^k$ for all $K$ nodes: $\boldsymbol{w}_{(t+1)}^k := \boldsymbol{w}_{(t)}^k - \frac{1}{K} \sum_{i=1}^K \boldsymbol{s}_{(t)}^k \boldsymbol{p}_{(t)}^k$.
17:     **end for**
18: **end for**

---

Table 15: Top-1 **test accuracy** of training **ResNet-20** on **CIFAR** ($KB_{\text{loc}} = 2048$ and $K = 16$).

|  | CIFAR-10 | CIFAR-100 |
|---|---|---|
| signSGD (average over signs) | 91.61 ±0.28 | 67.15 ±0.10 |
| signSGD (majority vote over signs) | 91.61 ±0.26 | 67.21 ±0.11 |

**Post-local SGD with EF-signSGD.** We extend the single-worker EF-signSGD algorithm of Karimireddy et al. (2019) (i.e. their Algorithm 1) to multiple-workers case by empirically investigating different algorithmic design choices. Our presented Algorithm 4 of EF-signSGD for multiple-workers can reach a similar performance as the mini-batch SGD counterpart (both are after the proper hyper-parameters tuning and under the same experimental setup).

**Training schemes, hyper-parameter tuning procedure for signSGD, signSGD variant in Algorithm 3, and EF-signSGD in Algorithm 4.** The training schemes of the signSGD (Bernstein et al., 2018) (i.e. the one using majority vote) and the signSGD variant in Algorithm 3 are slightly different from the the experimental setup mentioned in A.4, where we only fine-tune the initial learning rate in signSGD and Algorithm 3 (when $H = 1$). The optimal learning rate is searched from the grid $\{0.005, 0.10, 0.15, 0.20\}$ with the general fine-tuning principle mentioned in A.4. No learning rate

---

**Algorithm 4** *(Post-)Local SGD with the compression scheme in EF-signSGD*

---

**input:** the initial model $\boldsymbol{w}_{(0)} \in \mathbb{R}^d$; training data with labels $\mathcal{I}$; mini-batch of size $B_{\text{loc}}$ per local model; step size $\eta$, and momentum $\boldsymbol{m}$ (optional); number of synchronization steps $T$, and the first learning rate decay is performed at $T'$; number of eventual local steps $H'$; number of nodes $K$. error-compensated memory $\boldsymbol{e}_{(0)} = \boldsymbol{0} \in \mathbb{R}^d$.

1: synchronize to have the same initial models $\boldsymbol{w}_{(0)}^k := \boldsymbol{w}_{(0)}$.
2: initialize the local error memory $\boldsymbol{e}_{(0)}^k := \boldsymbol{e}_{(0)}$.
3: **for all** $k := 1, \ldots, K$ **do in parallel**
4:     **for** $t := 1, \ldots, T$ **do**
5:         **if** $t < T'$ **then**
6:             $H_{(t)} = 1$
7:         **else**
8:             $H_{(t)} = H'$
9:         **end if**
10:         **for** $h := 1, \ldots, H_{(t)}$ **do**
11:             sample a mini-batch from $\mathcal{I}_{(t)+h-1}^k$
12:             compute the gradient $\boldsymbol{g}_{(t)+h-1}^k := \frac{1}{B_{\text{loc}}} \sum_{i \in \mathcal{I}_{(t)+h-1}^k} \nabla f_i\left(\boldsymbol{w}_{(t)+h-1}^k\right)$.     $\triangleright$ can involve weight decay and momentum.
13:             update the local model $\boldsymbol{w}_{(t)+h}^k := \boldsymbol{w}_{(t)+h-1} - \gamma_{(t)} \boldsymbol{g}_{(t)+h-1}^k$.
14:         **end for**
15:         get model difference $\Delta_{(t)}^k := \boldsymbol{w}_{(t)}^k - \boldsymbol{w}_{(t)+H}^k + \boldsymbol{e}_{(t)}$.
16:         compress the model difference: $\boldsymbol{s}_{(t)}^k = \text{sign}(\Delta_{(t)}^k)$ and $\boldsymbol{p}_{(t)}^k = \frac{\left\|\Delta_{(t)}^k\right\|_1}{d}$.
17:         update local memory: $\boldsymbol{e}_{(t+1)}^k = \boldsymbol{e}_{(t)}^k - \boldsymbol{s}_{(t)}^k \boldsymbol{p}_{(t)}^k$.
18:         get new global (synchronized) model $\boldsymbol{w}_{(t+1)}^k$ for all $K$ nodes: $\boldsymbol{w}_{(t+1)}^k := \boldsymbol{w}_{(t)}^k - \frac{1}{K} \sum_{i=1}^K \boldsymbol{s}_{(t)}^k \boldsymbol{p}_{(t)}^k$.
19:     **end for**
20: **end for**

---

scaling up or warming up is required during the training, and the learning rate will be decayed by $10$ when accessing $50\%$ and $75\%$ of the total training samples.

The training schemes of the distributed EF-signSGD in Algorithm 4 in general follow the experimental setup mentioned in A.4. We grid search the optimal initial learning rate for Algorithm 4 (for the case of $H = 1$), and gradually warm up the learning rate from a relatively small value (0.1) to this found initial learning rate during the first 5 epochs of the training.

Note that in our experiments, weight decay and Nesterov momentum are used for the local model updates, for both of Algorithm 3 and Algorithm 4. We empirically found these two techniques can significantly improve the training/test performance under a fixed training epoch budget.

# D  Hierarchical Local SGD

The idea of local SGD can be leveraged to the more general setting of training on decentralized and heterogeneous systems, which is an increasingly important application area. Such systems have become common in the industry, e.g. with GPUs or other accelerators grouped hierarchically within machines, racks or even at the level of several data-centers. Hierarchical system architectures such as in Figure 17 motivate our hierarchical extension of local SGD. Moreover, end-user devices such as mobile phones form huge heterogeneous networks, where the benefits of efficient distributed and data-local training of machine learning models promises strong benefits in terms of data privacy.

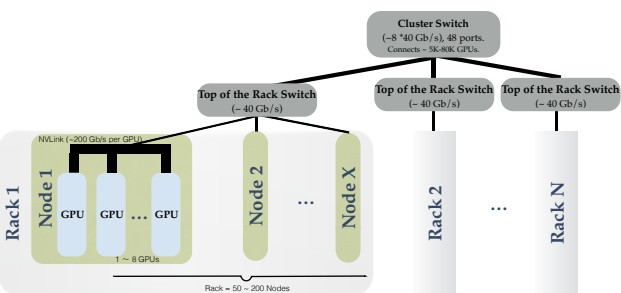

Figure 17: Illustration of a hierarchical network architecture of a cluster in the data center. While GPUs within each node are linked with fast connections (e.g. NVLink), connections between the servers within and between different racks have much lower bandwidth and latency (via top-of-the-rack switches and cluster switches). The hierarchy can be extended several layers further and further. Finally, edge switches face the external network at even lower bandwidth.

## D.1  The Illustration of Hierarchical Local SGD

Real world systems come with different communication bandwidths on several levels. In this scenario, we propose to employ local SGD on each level of the hierarchy, adapted to each corresponding computation vs communication trade-off. The resulting scheme, hierarchical local SGD, can offer significant benefits in system adaptivity and performance.

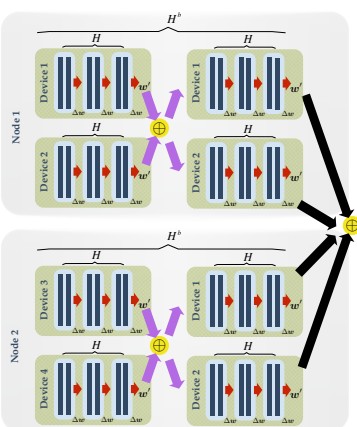

Figure 18: An illustration of hierarchical local SGD, for $B_{\mathrm{loc}} = 2$, using $H = 3$ inner local steps and $H^b = 2$ outer 'block' steps. Local parameter updates are depicted in red, whereas block and global synchronization is depicted in purple and black respectively.

As the guiding example, we consider compute clusters which typically allocate a large number of GPUs grouped over several machines, and refer to each group as a GPU-block. Hierarchical local SGD continuously updates the local models on each GPU for a number of $H$ *local update steps* before a (fast) synchronization within a GPU-block. On the outer level, after $H^b$ such *block update steps*, a

(slower) global synchronization over all GPU-blocks is performed. Figure 18 and Algorithm 5 depict how the hierarchical local SGD works, and the complete procedure is formalized below:

$$\boldsymbol{w}^k_{[(t)+l]+H} := \boldsymbol{w}^k_{[(t)+l]} - \sum_{h=1}^{H} \frac{\gamma_{[(t)]}}{B_{\text{loc}}} \cdot \sum_{i \in \mathcal{I}^k_{[(t)+l]+h-1}} \nabla f_i\big(\boldsymbol{w}^k_{[(t)+l]+h-1}\big)$$

$$\boldsymbol{w}^k_{[(t)+l+1]} := \boldsymbol{w}^k_{[(t)+l]} - \frac{1}{K_i} \sum_{k=1}^{K_i} \big(\boldsymbol{w}^k_{[(t)+l]} - \boldsymbol{w}^k_{[(t)+l]+H}\big)$$

$$\boldsymbol{w}^k_{[(t+1)]} := \boldsymbol{w}^k_{[(t)]} - \frac{1}{K} \sum_{k=1}^{K} \big(\boldsymbol{w}^k_{[(t)]} - \boldsymbol{w}^k_{[(t)+H^b]}\big) \tag{5}$$

where $\boldsymbol{w}^k_{[(t)+l]+H}$ indicates the model after $l$ block update steps and $H$ local update steps, and $K_i$ is the number of GPUs on the GPU-block $i$. The definition of $\gamma_{[(t)]}$ and $\mathcal{I}^k_{[(t)+l]+h-1}$ follows a similar scheme.

As the number of devices grows to the thousands (Goyal et al., 2017; You et al., 2017b), the difference between 'within' and 'between' block communication efficiency becomes more drastic. Thus, the performance benefits of our adaptive scheme compared to flat & large mini-batch SGD will be even more pronounced.

### D.2 THE ALGORITHM OF HIERARCHICAL LOCAL SGD

---

**Algorithm 5** *Hierarchical Local SGD*

---

**input:** the initial model $\boldsymbol{w}_{[(0)]}$;
**input:** training data with labels $\mathcal{I}$;
**input:** mini-batch of size $B_{\mathrm{loc}}$ per local model;
**input:** step size $\eta$, and momentum $m$ (optional);
**input:** number of synchronization steps $T$ over nodes;
**input:** number of local update steps $H$, and block update steps $H^b$;
**input:** number of nodes $K$ in total; and nodes $K'$ per GPU-block.

1: synchronize to have the same initial models $\boldsymbol{w}_{[(0)]}^{k} := \boldsymbol{w}_{[(0)]}$.
2: **for all** $k := 1, \ldots, K$ **do in parallel**
3:     **for** $t := 1, \ldots, T$ **do**
4:         **for** $l := 1, \ldots, H^b$ **do**
5:             **for** $h := 1, \ldots, H$ **do**
6:                 sample a mini-batch from $\mathcal{I}_{[(t)+l]+h-1}^{k}$.
7:                 compute the gradient

$$\boldsymbol{g}_{[(t)+l]+h-1}^{k} := \frac{1}{B_{\mathrm{loc}}} \sum_{i \in \mathcal{I}_{[(t)+l]+h-1}^{k}} \nabla f_i\left(\boldsymbol{w}_{[(t)+l]+h-1}^{k}\right).$$

8:                 update the local model

$$\boldsymbol{w}_{[(t)+l]+h}^{k} := \boldsymbol{w}_{[(t)+l]+h-1}^{k} - \gamma_{[(t)]} \boldsymbol{g}_{[(t)+l]+h-1}^{k}.$$

9:             **end for**
10:             inner all-reduce aggregation of the gradients

$$\Delta_{[(t)+l]}^{k} := \boldsymbol{w}_{[(t)+l]}^{k} - \boldsymbol{w}_{[(t)+l]+H}^{k}.$$

11:             get new block (synchronized) model $\boldsymbol{w}_{[(t)+l+1]}^{k}$ for $K'$ block nodes:

$$\boldsymbol{w}_{[(t)+l+1]}^{k} := \boldsymbol{w}_{[(t)+l]}^{k} - \gamma_{[(t)]} \tfrac{1}{K'} \sum_{k=1}^{K'} \Delta_{[(t)+l]}^{k},$$

12:         **end for**
13:         outer all-reduce aggregation of the gradients

$$\Delta_{[(t)]}^{k} := w_{[(t)]}^{k} - w_{[(t)+H^b]}^{k}.$$

14:         get new global (synchronized) model $\boldsymbol{w}_{[(t+1)]}^{k}$ for all $K$ nodes:

$$\boldsymbol{w}_{[(t+1)]}^{k} := \boldsymbol{w}_{[(t)]}^{k} - \gamma_{[(t)]} \tfrac{1}{K} \sum_{i=1}^{K} \Delta_{[(t)]}^{k}.$$

15:     **end for**
16: **end for**

---

### D.3 HIERARCHICAL LOCAL SGD TRAINING

Now we move to our proposed training scheme for distributed heterogeneous systems. In our experimental setup, we try to mimic the real world setting where several compute devices such as GPUs are grouped over different servers, and where network bandwidth (e.g. Ethernet) limits the communication of updates of large models. The investigation of hierarchical local SGD again trains ResNet-20 on CIFAR-10 and follows the same training procedure as local SGD where we re-formulate below.

The experiments follow the common mini-batch SGD training scheme for CIFAR (He et al., 2016a;b) and all competing methods access the same total amount of data samples regardless of the number of local steps or block steps. More precisely, the training procedure is terminated when the distributed algorithms have accessed the same number of samples as a standalone worker would access in 300 epochs. The data is partitioned among the GPUs and reshuffled globally every epoch. The local mini-batches are then sampled among the local data available on each GPU. The learning rate scheme is the same as in He et al. (2016a), where the initial learning rate starts from $0.1$ and is divided by $10$ when the model has accessed $50\%$ and $75\%$ of the total number of training samples. In addition to this, the momentum parameter is set to $0.9$ without dampening and applied independently to each local model.

#### D.3.1 THE PERFORMANCE OF HIERARCHICAL LOCAL SGD.

Table 16: Training **CIFAR-10** with **ResNet-20** via **local SGD** on a $8 \times 2$-GPU cluster. The local batch size $B_{\text{loc}}$ is fixed to 128 with $H^b = 1$, and we scale the number of local steps $H$ from 1 to 1024. The reported **training times** are the average of three runs and all the experiments are under the same training configurations for the equivalent of 300 epochs, without specific tuning.

| $H =$ | 1 | 2 | 4 | 8 | 16 | 32 | 64 | 128 | 256 | 512 | 1024 |
|---|---|---|---|---|---|---|---|---|---|---|---|
| Training Time (minutes) | 20.07 | 13.95 | 10.48 | 9.20 | 8.57 | 8.32 | 9.22 | 9.23 | 9.50 | 10.30 | 10.65 |

**Training time vs. local number of steps.** Table 16 shows the performance of local SGD in terms of training time. The communication traffic comes from the global synchronization over 8 nodes, each having 2 GPUs. We can witness that increasing the number of local steps over the "datacenter" scenario cannot infinitely improve the communication performance, or would even reduce the communication benefits brought by a large number of local steps. *Hierarchical local SGD with inner node synchronization reduces the difficulty of synchronizing over the complex heterogeneous environment, and hence enhances the overall system performance of the synchronization. The benefits are further pronounced when scaling up the cluster size.*

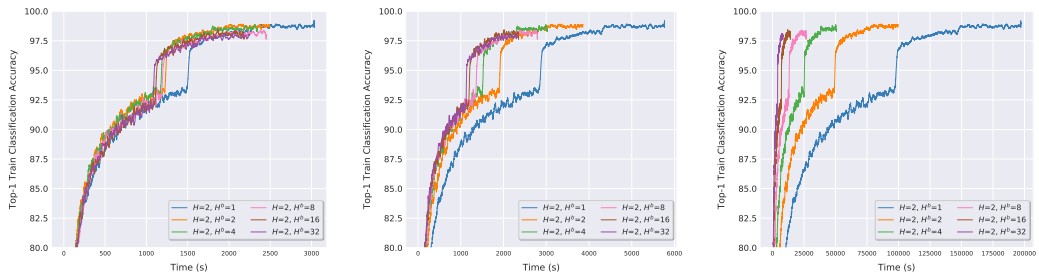

(a) Training accuracy vs. time. The number of local steps is $H = 2$.

(b) Training accuracy vs. time. The number of local steps is $H = 2$, with 1 second delay for each global synchronization.

(c) Training accuracy vs. time. The number of local steps is $H = 2$, with 50 seconds delay for each global synchronization.

Figure 19: The performance of **hierarchical local SGD** trained on **CIFAR-10** with **ResNet-20** ($2 \times 2$-GPU). Each GPU block of the hierarchical local SGD has 2 GPUs, and we have 2 blocks in total. Each figure fixes the number of local steps but varies the number of block steps from 1 to 32. All the experiments are under the same training configurations without specific tuning.

**Hierarchical local SGD shows high tolerance to network delays.** Even in our small-scale experiment of two servers and each with two GPUs, *hierarchical local SGD shows its ability to significantly reduce the communication cost by increasing the number of block step $H^b$ (for a fixed $H$), with trivial performance degradation. Moreover, hierarchical local SGD with a sufficient number of block steps offers strong robustness to network delays.* For example, for fixed $H = 2$, by increasing the number of $H^b$, i.e. reducing the number of global synchronizations over all models, we obtain a significant gain in training time as in Figure 19(a). The impact of a network of slower communication is further studied in Figure 19(b), where the training is simulated in a realistic scenario and each global communication round comes with an additional delay of 1 second. Surprisingly, even for the global synchronization with straggling workers and severe $50$ seconds delay per global communication round, Figure 19(c) demonstrates that a large number of block steps (e.g. $H^b = 16$) still manages to fully overcome the communication bottleneck with no/trivial performance damage.

Table 17: The performance of training **CIFAR-10** with **ResNet-20** via **hierarchical local SGD** on a 16-GPU Kubernetes cluster. We simulate three different types of cluster topology, namely 8 nodes with 2 GPUs/node, 4 nodes with 4 GPUs/node, and 2 nodes with 8 GPUs/node. The configuration of hierarchical local SGD satisfies $H \cdot H^b = 16$. All variants either synchronize within each node or over all GPUs, and the communication cost is estimated by only considering $H \cdot H^b = 16$ model updates during the training (the update could come from a different level of the synchronizations). The reported results are the average of three runs and all the experiments are under the same training configurations, training for the equivalent of 300 epochs, without specific tuning.

| | $H = 1,$ $H^b = 16$ | $H = 2,$ $H^b = 8$ | $H = 4,$ $H^b = 4$ | $H = 8,$ $H^b = 2$ | $H = 16,$ $H^b = 1$ |
|---|---|---|---|---|---|
| # of sync. over nodes | 1 | 1 | 1 | 1 | 1 |
| # of sync. within node | 15 | 7 | 3 | 1 | 0 |
| Test acc. on $8 \times 2$-GPU | 90.02 $\pm 0.28$ | 90.25 $\pm 0.08$ | 89.95 $\pm 0.19$ | 91.41 $\pm 0.23$ | 91.18 $\pm 0.02$ |
| Test acc. on $4 \times 4$-GPU | 91.65 $\pm 0.06$ | 91.26 $\pm 0.17$ | 91.46 $\pm 0.24$ | 91.91 $\pm 0.16$ | |
| Test acc. on $2 \times 8$-GPU | 92.14 $\pm 0.10$ | 92.05 $\pm 0.14$ | 91.94 $\pm 0.09$ | 91.56 $\pm 0.18$ | |

**Hierarchical local SGD offers improved scaling and better test accuracy.** Table 17 compares the mini-batch SGD with hierarchical local SGD for fixed product $H \cdot H^b = 16$ under different network topologies, with the same training procedure. We can observe that for a heterogeneous system with a sufficient block size, hierarchical local SGD with a sufficient number of block update steps can further improve the generalization performance of local SGD training. More precisely, when $H \cdot H^b$ is fixed, hierarchical local SGD with more frequent inner-node synchronizations ($H^b > 1$) outperforms local SGD ($H^b = 1$), while still maintaining the benefits of significantly reduced communication by the inner synchronizations within each node. In summary, as witnessed by Table 17, *hierarchical local SGD outperforms both local SGD and mini-batch SGD in terms of training speed as well as model performance*, especially for the training across nodes where inter-node connection is slow but intra-node communication is more efficient.

## E    COMMUNICATION SCHEMES

This section evaluates the communication cost in terms of the number of local steps and block steps, and formalizes the whole communication problem below.

Assume $K$ computing devices uniformly distributed over $K'$ servers, where each server has $\frac{K}{K'}$ devices. The hierarchical local SGD training procedure will access $N$ total samples with local mini-batch size $B$, with $H$ local steps and $H^b$ block steps.

The MPI communication scheme (Gropp et al., 1999) is introduced for communication cost evaluation. More precisely, we use general all-reduce, e.g., *recursive halving and doubling algorithm* (Thakur et al., 2005; Rabenseifner, 2004), for gradient aggregation among $K$ computation devices. For each all-reduce communication, it introduces $C \cdot \log_2 K$ communication cost, where $C$ is the message transmission time plus network latency.

The communication cost under our hierarchical local SGD setting is mainly determined by the number of local steps and block steps. The $\frac{K}{T}$ models within each server synchronize the gradients for every $H$ local mini-batch, and it only performs global gradients aggregation of $K$ local models after $H^b$ block updates. Thus, the total number of synchronizations among compute devices is reduced to $\lceil \frac{N}{KB \cdot HH^b} \rceil$, and we can formulate the overall communication cost $\tilde{C}$ as:

$$\tilde{C} \approx \left( \lceil \tfrac{N}{KB \cdot H} \rceil - \lceil \tfrac{N}{KB \cdot HH^b} \rceil \right) \cdot C_1 \cdot K' \log_2 \tfrac{K}{K'} + \lceil \tfrac{N}{KB \cdot HH^b} \rceil \cdot C_2 \log_2 K \qquad (6)$$

where $C_1$ is the single message passing cost for compute devices within the same server, $C_2$ is the cost of that across servers, and obviously $C_1 \ll C_2$. We can easily witness that the number of block steps $H^b$ is more deterministic in terms of communication reduction than local step $H$. Empirical evaluations can be found in Section D.3.

Also, note that our hierarchical local SGD is orthogonal to the implementation of gradient aggregation (Goyal et al., 2017) optimized for the hardware, but focusing on overcoming the aggregation cost of more general distributed scenarios, and can easily be integrated with any optimized all-reduce implementation.

## F  DISCUSSION AND FUTURE WORK

**Data distribution patterns.**    In our experiments, the dataset is globally shuffled once per epoch and each local worker only accesses a disjoint part of the training data. Removing shuffling altogether, and instead keeping the disjoint data parts completely local during training might be envisioned for extremely large datasets which can not be shared, or also in a federated scenario where data locality is a must for privacy reasons. This scenario is not covered by the current theoretical understanding of local SGD, but will be interesting to investigate theoretically and practically.

**Better learning rate scheduler for local SGD.**    We have shown in our experiments that local SGD delivers consistent and significant improvements over the state-of-the-art performance of mini-batch SGD. For ImageNet, we simply applied the same configuration of "large-batch learning schemes" by Goyal et al. (2017). However, this set of schemes was specifically developed and tuned for mini-batch SGD only, not for local SGD. For example, scaling the learning rate w.r.t. the global mini-batch size ignores the frequent local updates where each local model only accesses local mini-batches for most of the time. Therefore, it is expected that specifically deriving and tuning a learning rate scheduler for local SGD would lead to even more drastic improvements over mini-batch SGD, especially on larger tasks such as ImageNet.

**Adaptive local SGD.**    As local SGD achieves better generalization than current mini-batch SGD approaches, an interesting question is if the number of local steps $H$ could be chosen adaptively, i.e. change during the training phase. This could potentially eliminate or at least simplify complex learning rate schedules. Furthermore, recent works (Loshchilov & Hutter, 2017; Huang et al., 2017a) leverage cyclic learning rate schedules either improving the general performance of deep neural network training, or using an ensemble multiple neural networks at no additional training cost. Adaptive local SGD could potentially achieve similar goals with reduced training cost.

**Hierarchical local SGD design.**    Hierarchical local SGD provides a simple but efficient training solution for devices over the complex heterogeneous system. However, its performance might be impacted by the cluster topology. For example, the topology of $8 \times 2$-GPU in Table 17 fails to further improve the performance of local SGD by using more frequent inner node synchronizations. On contrary, sufficiently large size of the GPU block could easily benefit from the block update of hierarchical local SGD, for both communication efficiency and training quality. The design space of hierarchical local SGD for different cluster topologies should be further investigated, e.g., to investigate the two levels of model averaging frequency (within and between blocks) in terms of convergence, and the interplay of different local minima in the case of very large number of local steps. Another interesting line of work, explores heterogenous systems by allowing for different number of local steps on different clusters, thus making up for slower machines.

