# OpenReview forum: "Don't Use Large Mini-batches, Use Local SGD"
_ICLR.cc/2020/Conference — Accept (Poster)_

### Official Review · AnonReviewer1 · 2019-10-23
**Official Blind Review #1**

**Rating:** 6

**Review:**

In this paper, the authors propose a variant of local SGD: post-local SGD, which improves the generalization performance compared to large-batch SGD. This paper also empirically studies the trade-off between communication efficiency and performance. Additionally, this paper proposes hierarchical local SGD. The paper is well-written, the experiments show good performance.

However, there are several weakness in this paper:

1. The post-local SGD is a simple extension of local SGD. Roughly speaking, post-local SGD uses fully synchronous SGD to warm up local SGD. The novelty of this algorithm is limited.

2. The main statement that post-local SGD improves the generalization performance, is only supported by empirical results. No theoretical analysis is provided. Thus, the contribution of this paper is limited.

3. In this paper, it is reported that in some experiments, local SGD significantly outperforms mini-batch SGD. As shown in Figure 3, when the number of workers is large enough, mini-batch SGD has extremely bad performance. However, such bad result is potentially caused by a bad choice of learning rates. In this paper, the authors use "linearly scaling the learning rate w.r.t. the global mini-batch size". However, some recent papers also suggest using square root scaling instead of linear scaling [1]. I think the mini-batch SGD fails simply because the learning rates are too large. However, the authors claim that the learning rates for mini-batch SGD are fine-tuned (in Figure 3), which makes the empirical results questionable.


-----------
References

[1] You, Yang, et al. "Large batch optimization for deep learning: Training bert in 76 minutes." arXiv preprint arXiv:1904.00962 (2019).

===================
Update after author's feedback:

I do not have strong reasons to doubt the empirical results after reading the author's feedback, so I increase the score.

**Experience Assessment:**

I have read many papers in this area.

**Review Assessment: Checking Correctness Of Derivations And Theory:**

N/A

**Review Assessment: Checking Correctness Of Experiments:**

I assessed the sensibility of the experiments.

**Review Assessment: Thoroughness In Paper Reading:**

I read the paper at least twice and used my best judgement in assessing the paper.

---

> ### Author Response · Authors · 2019-11-12
> **Response to Reviewer1**
>
> Thank you for your review. Please check our responses below.
>
> [We correctly finetune the learning rate for mini-batch SGD]
> We would like to clarify that our choices of learning rates are in favor of the baselines, not our method. We have carefully fine-tuned the learning rate for all mini-batch SGD baselines, while we rely on the ‘linear scaling rule’ for (post)-local SGD. As mentioned in the paragraph ‘The procedure of fine-tuning’ of Appendix A.4.1, instead of using ‘linear scaling’ [2] or ‘square root scaling’ [1] to automatically determine the scaling factor, we did perform grid search for the optimal scaling factor for all mini-batch SGD baselines (this grid search comprises linear scaling and square root scaling as special cases).
>
> We believe that these clarifications give sufficient evidence to show that your concern raised on the “questionable empirical results”, and specifically the learning rates in mini-batch SGD, is without cause. As this was one of your main concerns in your review, we would therefore like to ask you if you could reconsider your score?
>
> [Novelty of Algorithms]
> Reviewers 2 and 3 seem to perceive the simplicity and ease-of-use as an advantage of our method. We show that these known algorithmic building blocks, if combined correctly, can lead to remarkable improvements in generalization error, and at the same time alleviate problems in large batch training, and provide a comprehensive empirical study of the involved trade-offs.
>
> [Theoretical analysis of generalization performance]
> We agree that a theoretical proof of improved generalization ability of Local SGD variants over large batch SGD would be an impressive result. However, we are not aware of any such results even in the single machine case. Generalization results are typically much harder to prove than optimization results in the non-convex setting. The analysis would have to depend on batch size and learning rate, in order to be useful for the large batch setting. And most results which do, don’t tell anything useful about the generalization of deep neural networks. Thus we believe that at this point, we can only talk about intuitive explanations (such as injecting noise) as opposed to “why Local SGD works?” (or large-batch doesn’t).
>
> Moreover, our work can provide new insights for the theoretical investigation of the local SGD. Please check our first item in our response to Reviewer2.
>
>
> [2] Goyal, et al. "Accurate, large minibatch SGD: Training imagenet in 1 hour." arXiv preprint arXiv:1706.02677 (2017).

---

> > ### Comment · AnonReviewer1 · 2019-11-13
> > **Author's feedback sounds reasonable**
> >
> > Since the other reviewers all give positive reviews, and I do not have strong reasons to doubt the empirical results after reading the author's feedback, I increase the score.

---

### Official Review · AnonReviewer3 · 2019-10-23
**Official Blind Review #3**

**Rating:** 6

**Review:**

This paper proposes a variant of local SGD, post-local SGD, for distributed training of deep neural networks. It targets to mitigate the generalization gap caused by large batch training. The idea is straightforward and easy to understand-- start the training with standard mini-batch SGD and later switch to local SGD. The rationale behind this scheme is that switching to local SGD helps the training converge to flatter minima compared to using large-batch SGD, which correlates with sharper minima, and that helps close the generation gap. Switching to local SGD at the second phase also helps improve communication efficiency by reducing the amortized communication volume. The authors perform empirical studies using ResNet and DenseNet to conclude that post-local SGD outperforms large-batch SGD in terms of generalization performance while also with improved communication efficiency.

Strengths:
+ The post-local SGD technique is simple yet seems to be useful in practice.
+ Provide a thorough evaluation of the communication efficiency and generalization performance of local SGD.
+ Introduce a hierarchical version of post-local SGD to better adapt to the topology of the GPU cluster, which often consists of heterogeneous interconnects.

Weaknesses:
- The design and experiments are largely empirical without theoretical derivation.
- It is less clear about the benefit of post-local SGD when applied to ADAM, which is widely used for distributed training of NLP tasks.
- Scalability improvements over mini-batch SGD are largely done by ignoring other optimization techniques that also reduce the communication volume, such as gradient compression[1], Terngrad[2].

Overall, the post-local SGD proposed by the paper seems to be a promising technique for large-scale distributed training. The motivation and explanation of the work are clear. My major concern is about the generalizability of this work. ResNet50 is not that interesting from a distributed training perspective. It is less clear whether the performance gains are consistent across tasks. The authors are encouraged to report experimental results on distributed training of large LM models.

[1]"Deep Gradient Compression: Reducing the Communication Bandwidth for Distributed Training", Lin et. al., ICLR 2018
[2]"Terngrad: Ternary gradients to reduce communication in distributed deep learning", Wen et. al., NeurIPS 2017

**Experience Assessment:**

I have published one or two papers in this area.

**Review Assessment: Checking Correctness Of Derivations And Theory:**

I assessed the sensibility of the derivations and theory.

**Review Assessment: Checking Correctness Of Experiments:**

I assessed the sensibility of the experiments.

**Review Assessment: Thoroughness In Paper Reading:**

I read the paper thoroughly.

---

> ### Author Response · Authors · 2019-11-12
> **Response to Reviewer3**
>
> Thank you for your review. We’ve updated the draft and answer your concerns below:
>
> [Scalability, and generalize post-local SGD with other compression techniques]
> About scalability improvements over mini-batch SGD, Figure 2 does in fact show the transition between fast communication to the setting when communication does become a limiting factor, by giving complete time-to-accuracy numbers including communication.
>
> Gradient compression techniques are indeed orthogonal to our approach. In contrast to gradient compression, local SGD does change the trade-off between the number of communications and back-propagations, which can be crucial in high latency scenarios. Nevertheless, in the updated submission, we include a table (Table 3) of new results in Section 4.2 to demonstrate the compatibility of post-local SGD w.r.t. this aspect. Post-local SGD can be combined with the SOTA sign-based compression techniques [3, 4], not only addressing the quality loss issues introduced by the compressed communication, but also further improve the scalability. We will try to add more results to this aspect.
>
> [Experiments on language models]
> Some preliminary results for training 3-layer LSTMs for language modeling on WikiText2 are included in Table 12 (Appendix C.5.2), where we show that post-local SGD can still improve large-batch SGD without any hyper-parameter tuning.
>
> Generalizing post-local SGD to Adam for other NLP tasks is interesting but out of the scope of the current paper. We focus on addressing the generalization issue of deep learning, where mini-batch SGD with momentum is the SOTA optimizer and adaptive methods have not been equally successful for generalization [5].
>
> [The theoretical contributions]
> Please check our response to Reviewer1 and Reviewer2 for more details.
>
>
> [3] Bernstein, et al. "signSGD: Compressed optimisation for non-convex problems." ICML, 2018.
> [4] Karimireddy, et al. "Error feedback fixes signSGD and other gradient compression schemes." ICML, 2019.
> [5] Wilson, et al. "The marginal value of adaptive gradient methods in machine learning." Advances in Neural Information Processing Systems. 2017.

---

### Official Review · AnonReviewer2 · 2019-10-24
**Official Blind Review #2**

**Rating:** 6

**Review:**

This paper proposes a new distributed computation technique for SGD training of deep neural networks. The proposed method is a modification of the local SGD which updates models distributed to several workers in a parallel way and synchronize the model parameters at every few epochs. The local SGD shows a nice performance but it is not robust against large mini-batch size. The proposed method is called post-local SGD that starts the local SGD after some epochs of standard mini-batch SGD. This modification makes the local SGD robust against the large mini-batch size. The authors conducted thorough experiments to investigate the performance of the proposed method. The experiments reveal that the proposed method gives better performances than the mini-batch SGD and the vanilla local SGD.

Pros:
- As far as I checked, the numerical experiments are strong. They checked several points of view. Several settings of mini-batch sizes and number of workers are compared. For more detailed points, they compared the proposed method for not only the vanilla SGD but also other optimizers such as momentum SGD. Moreover, different choices of timing of starting local SGD (t') are compared.
- The proposed method is simple and easy to implement.

Cons:
- The local SGD itself is not new and has been already proposed (indeed, the authors are also explaining this point in the paper), and theoretical investigation of local SGD has also been well exploited. This method is just a combination of normal mini-batch SGD and local SGD. In that sense, there are not a so much eye-opening idea in the proposed method.
- SGLD interpretation is instructive, but it does not explain why the "post"-local SGD outperforms the local SGD. We can intuitively guess why post-local SGD is better but it is not a rigorous theory.
- The post local SGD algorithm has more freedom than existing method, which results in more tuning parameters. There are parameters such as the batch-size before and after switching, the number of workers, the timing of switching and so on. It could be disadvantage of the proposed method.

Minor comment:
- Please give the precise meaning of the dominant eigenvalue.
- I think the setting that the data are distributed to all workers is acceptable. It is also an interesting setting from the HPC point of view (on the other hand, it would be less interesting in terms of federated learning).

**Experience Assessment:**

I have read many papers in this area.

**Review Assessment: Checking Correctness Of Derivations And Theory:**

I assessed the sensibility of the derivations and theory.

**Review Assessment: Checking Correctness Of Experiments:**

I assessed the sensibility of the experiments.

**Review Assessment: Thoroughness In Paper Reading:**

I read the paper at least twice and used my best judgement in assessing the paper.

---

> ### Author Response · Authors · 2019-11-12
> **Response to Reviewer2**
>
> Thank you for your review. We answer your specific questions below.
>
> [New insights for theoretical investigation of local SGD]
> The established theoretical work for local SGD considers convex loss or non-convex training loss, but none of these work discusses the generalization issues. Even the prior theoretical work cannot explain the failure of large-batch SGD for non-convex deep learning.
>
> In that light, it is a far-fetched claim that Local SGD is well understood in the non-convex case. In fact, compared to the local SGD variant in [1] who borrowed the theory from the optimization analysis, post-local SGD is an opposite algorithm based on the understanding of generalization in deep learning.
>
> Our empirical experiments and superior performance provide new valuable insights for the community to better understand the convergence analysis and deep learning generalization theory.
>
> [Post-local SGD is a simple plugin w/o involving heavy tuning]
> We performed extensive ablation studies and provided practical guidelines for post-local SGD.
> Our algorithm does not have any additional hyperparameters when compared to local SGD. As in local SGD, the only hyper-parameter which needs to be tuned is the number of local update steps H, which (e.g. illustrated in Figure 4) determines (1) how much better the post-local SGD can be over the mini-batch SGD, and (2) how much communication cost can be reduced.
>
> Below we summarize the detailed evidence to justify the effectiveness of post-local SGD.
> 1. In all our evaluations, we fixed the local batch size B_loc to the value reported in the literature.
> 2. Figure 4 studies how different numbers of workers, and different numbers of local update steps, will impact the generalization performance. Post-local SGD in general improves over mini-batch SGD.
> 3. We provide extensive ablation studies on the switching time of post-local SGD in the Appendix (Figures 11, 12, 13). For our post-local SGD evaluations, we turn on post-local SGD after the first learning rate decay, for better generalization performance and communication efficiency.
>
> [The meaning of the dominant eigenvalue of the Hessian]
> The eigenvalues of the Hessian of the loss characterize the local curvature of the loss. Some techniques are developed to better understand the landscape of the loss surface, as well as the generalization properties [2, 3, 4]. Borrowing the explanation from [3]:  “the spectrum of the Hessian is composed of two parts: (1) the bulk centered near zero, (2) and outliers away from the bulk which appears to depend on the data”. It implies most directions in the weight space are flat, and leads to little or no change in the loss value, except for the directions of eigenvectors that correspond to the large eigenvalues of the Hessian.
>
> For more details, please check [3, 4].
>
>
> [1] Wang, et al. "Adaptive communication strategies to achieve the best error-runtime trade-off in local-update SGD." SysML, 2019.
> [2] Yao, et al. "Hessian-based analysis of large batch training and robustness to adversaries." NeurIPS. 2018.
> [3] Sagun, et al. "Empirical analysis of the Hessian of over-parametrized neural networks." ICLR workshop, 2018.
> [4] Ghorbani, et al. "An investigation into neural net optimization via hessian eigenvalue density." ICML 2019.

---

### Public Comment · ~Stone_Jamess1 · 2019-10-15
**Use distributed deep learning is misleading**

I think the author uses the distributed deep learning is misleading.

In the experiment part, the author say " The data is disjointly partitioned and reshuffled globally every epoch. ". So it violates  the setting in most distributed deep learning that data is kept locally at each node.
Consider the case, if the data is shuffled many times, then each node will get access to the full dataset, so it weakens the experimental part. In fact, in many real applications, it is challenging and time consuming to reshuffle the data again and again.

---

> ### Author Response · Authors · 2019-10-16
> **We believe the experimental setup is standard and correct**
>
> Thank you for your interest. The setting in which data is kept locally to the workers is normally called 'decentralized'. Note that the ‘distributed’ setting does not always have to be ‘decentralized’.
>
> We consider the 'distributed' (data center) setting where each worker has fast access to all data points. This is a standard setup and very important in deep learning. See for instance [1], the current state-of-the-art baseline for large-batch training in the datacenter.
>
> We use the formulation 'disjointly partitioned' to indicate that data points are used only once (across workers) in each epoch. To achieve this, we don't reshuffle the data points themselves. We just shuffle the data indices with a shared random seed across workers at the beginning of each epoch. Each worker then takes a disjoint part of the data indices to generate local mini-batches.
>
> We hope this resolves your concern.
>
> ------
> [1] Accurate, Large Minibatch SGD: Training ImageNet in 1 Hour, https://arxiv.org/abs/1706.02677

---

> > ### Comment · AnonReviewer1 · 2019-10-16
> > **Experimental setup is correct, though the IID setting is a little bit weak.**
> >
> > First, I agree with the authors that the experimental setup is standard and correct.
> > Basically, each epoch is a full pass of the entire dataset, and the same data sample won't be processed twice in the same epoch.
> >
> > This paper takes the assumption that each worker has the access to the entire dataset.
> >
> > However, typically, decentralized/non-IID setting is considered more difficult.
> > Furthermore, when the dataset is extremely large, it is reasonable to disjointly partition the dataset to the workers and never reshuffle and re-partition it.
> > Note that the previous comment never claims that the experimental setup is incorrect.
> > And, since there are other papers of local SGD [2,3] that assume decentralized/non-IID local datasets on the workers, the contribution of this paper is potentially weakened.
> >
> > ----------
> > [2] Yu, Hao, Sen Yang, and Shenghuo Zhu. "Parallel restarted SGD with faster convergence and less communication: Demystifying why model averaging works for deep learning." Proceedings of the AAAI Conference on Artificial Intelligence. Vol. 33. 2019.
> > [3] Yu, Hao, Rong Jin, and Sen Yang. "On the Linear Speedup Analysis of Communication Efficient Momentum SGD for Distributed Non-Convex Optimization." International Conference on Machine Learning. 2019.

---

> > > ### Public Comment · ~Stone_Jamess1 · 2019-10-16
> > > **Thanks for your replies**
> > >
> > > Thanks for all of your responses.
> > > I agree that the approach cannot be used for distributed mobile users, so it weakens the contribution.
> > >
> > > In addition, I have another concern that each node has access to the full dataset and you just need to shuffle the index. Consider the following case, in every iteration, each user sample a batch of data from the entire dataset and performs the computation. I think it will make the computation more faster since each user doesn't need to be confined for a disjoint subset. From this point, I don't see the benefit of the disjoint setting.
> > > Furthermore, when we do the mini-batch setting for single GPU training, we also don't have the constraint each sample cannot be used twice since the mini-batch is randomly selected.
> > >
> > > Thanks again for all of your response.

---

> > > > ### Comment · AnonReviewer1 · 2019-10-16
> > > > **In practice, random permutation usually gives better convergence**
> > > >
> > > > In practice, it is observed that random permutation/random sampling without replacement usually converges faster than sampling with replacement [4] (there should be some better references but I don't remember...).
> > > > That's why we usually use random permutation in experiments. If you are familiar with the dataloader of pytorch, you will see that it is also implemented by random permutation.
> > > >
> > > > BTW, I don't really understand why " each user sample a batch of data from the entire dataset and performs the computation. I think it will make the computation more faster since each user doesn't need to be confined for a disjoint subset."
> > > > Here you mean "sampling with replacement is faster than sampling without replacement in computation"? why is that true?
> > > >
> > > >
> > > > ------
> > > > [4] Ying, Bicheng, et al. "On the performance of random reshuffling in stochastic learning." 2017 Information Theory and Applications Workshop (ITA). IEEE, 2017.

---

> > > > > ### Public Comment · ~Stone_Jamess1 · 2019-10-16
> > > > > **Thanks for your reply**
> > > > >
> > > > > Thanks for your quick response.
> > > > >
> > > > > The goal of this paper is to reduce the communication cost, so it will not aggregate the gradients for every iteration. So for each node, it can only get information from other data through the gradient aggregation.
> > > > >
> > > > > Since each user can get access to other data, why we confine each node reshuffle the data every epoch? The setting here allows each node get access to all data, then it is no necessary for other nodes to get information from gradients only.  It is my understanding from the setting of this paper.
> > > > >
> > > > > Thanks.

---

> > > > > > ### Comment · AnonReviewer1 · 2019-10-16
> > > > > > **Reshuffling is just a way to implement random sampling without replacement**
> > > > > >
> > > > > > As I mentioned, in practice, random sampling without replacement is preferred.
> > > > > > Reshuffling the indices at the beginning of each epoch and then sequentially load the data samples according to the shuffled indices is the easiest way to implement sampling without replacement.
> > > > > >
> > > > > > Since it is "without replacement", the same data sample should not be processed/sampled twice by different workers in the same epoch.
> > > > > > In a single epoch, although a worker have access to the data of the other workers, it won't use it.
> > > > > > Otherwise, there will be some extra computation.
> > > > > >
> > > > > > Another point of view is to make a fair comparison between local SGD and fully synchronous SGD. If we get the random seeds fixed, the workers in these 2 algorithms should process the same sequences of data. The only difference is that local SGD skips some rounds of synchronization.
> > > > > > Yes, it is possible for different workers to share some data in the same epoch in local SGD, however, that will make it unfair when compared to fully synchronous SGD.

---

> > > ### Author Response · Authors · 2019-10-16
> > > **Federated learning is beyond the scope of this paper**
> > >
> > > Thank you for the thoughtful comments. We agree that the decentralized/non-IID setting is important and much more difficult.
> > >
> > > The local SGD algorithm in that setting is also known as federated averaging [FA], and there is a very extensive recent literature on it, including the two papers you mention. The federated learning setting is beyond the scope of our paper here as it will introduce additional generalization challenges beyond the large-batch scenario of our interest here. Nevertheless we'll include both mentioned papers in the revised version.
> > >
> > > [FA]  McMahan, B., Moore, E., Ramage, D., Hampson, S., & y Arcas, B. A. (2017). Communication-Efficient Learning of Deep Networks from Decentralized Data. AISTATS 2017

---

> > > > ### Comment · AnonReviewer1 · 2019-10-16
> > > > **Federated learning is different from local SGD**
> > > >
> > > > I agree that federated learning (FL) is beyond the scope of this paper.
> > > >
> > > > However, I think [2,3] are local SGD, not FL.
> > > >
> > > > In my opinion, compared to local SGD, FL has the following major differences (and difficulties in theoretical analysis):
> > > > 1. subsampled workers: only a subset of workers participates the training in each epoch
> > > > 2. heterogeneous number of local steps: in local SGD, the number of local steps (H in your paper) must be the same across different workers, so that we can use an auxiliary (fake-average) variable to analyze the convergence. However, for federated learning, different workers can have different H, which makes it difficult for theoretical analysis.
> > > >
> > > > Decentralized/non-IID data is just another standard setting of traditional distributed SGD.
> > > > Non-IID data itself does not make [2,3] FL.

---

> > > > > ### Author Response · Authors · 2019-10-18
> > > > > **Orthogonal contributions**
> > > > >
> > > > > We acknowledge the theoretical contribution of the work [2,3] and will add it to the related work - however the contribution of our empirical paper is completely orthogonal to them. We focus on the generalization issue for *large-batch training*, an issue which is still unsolved even for iid data but of strong importance for distributed deep learning. We believe our contribution is significant in this aspect. [2,3] primarily focus on the optimization aspect in terms of theoretical insights. The global mini-batch size in their experiments is up to 512 for CIFAR-10 and 1024 for ImageNet; while ours reaches 4096 for CIFAR and 16K for ImageNet, focusing on the generalization challenge due to large mini-batch sizes.
> > > > >
> > > > > On a side-note, we would like to clarify that only [3] proves a convergence rate for both the iid and non-iid setup. The convergence proof in [2] relies on bounded gradients (Assumption 1) and iid data (Assumption 2) with might be difficult to generalize to the non-iid setting, as argued in a recent preprint [5].
> > > > >
> > > > > [5] A. Khaled, K. Mishchenko, P. Richtárik, First Analysis of Local GD on Heterogeneous Data, arxiv.org/abs/1909.04715

---

### Decision · Program_Chairs · 2019-12-19

**Decision:**

Accept (Poster)

**Comment:**

The authors propose a simple modification of local SGD for parallel training, starting with standard SGD and then switching to local SGD. The resulting method provides good results and makes a practical contribution. Please carefully account for reviewer comments in future revisions.